# Cached Multi-Lora Composition for Multi-Concept Image Generation

**Xiandong Zou, Mingzhu Shen**,[*] **Christos-Savvas Bouganis, Yiren Zhao**
Imperial College London, UK
{xiandong.zou20, m.shen23, christos-savvas.bouganis, a.zhao}@imperial.ac.uk

## Abstract

Low-Rank Adaptation (LoRA) has emerged as a widely adopted technique in text-to-image models, enabling precise rendering of multiple distinct elements, such as characters and styles, in multi-concept image generation. However, current approaches face significant challenges when composing these LoRAs for multi-concept image generation, particularly as the number of LoRAs increases, resulting in diminished generated image quality. In this paper, we initially investigate the role of LoRAs in the denoising process through the lens of the Fourier frequency domain. Based on the hypothesis that applying multiple LoRAs could lead to "semantic conflicts", we have conducted empirical experiments and find that certain LoRAs amplify high-frequency features such as edges and textures, whereas others mainly focus on low-frequency elements, including the overall structure and smooth color gradients. Building on these insights, we devise a frequency domain based sequencing strategy to determine the optimal order in which LoRAs should be integrated during inference. This strategy offers a methodical and generalizable solution compared to the naive integration commonly found in existing LoRA fusion techniques. To fully leverage our proposed LoRA order sequence determination method in multi-LoRA composition tasks, we introduce a novel, training-free framework, Cached Multi-LoRA (CMLoRA), designed to efficiently integrate multiple LoRAs while maintaining cohesive image generation. With its flexible backbone for multi-LoRA fusion and a non-uniform caching strategy tailored to individual LoRAs, CMLoRA has the potential to reduce semantic conflicts in LoRA composition and improve computational efficiency. Our experimental evaluations demonstrate that CMLoRA outperforms state-of-the-art training-free LoRA fusion methods by a significant margin – it achieves an average improvement of 2.19% in CLIPScore, and 11.25% in MLLM win rate compared to LoraHub, LoRA Composite, and LoRA Switch.[1]

## 1 Introduction

In the realm of generative text-to-image models (Ramesh et al., 2021; Saharia et al., 2022; Podell et al., 2023; Esser et al., 2024; Rombach et al., 2022; Ramesh et al., 2022), the integration of Low-Rank Adaptation (LoRA) (Hu et al., 2021) in image generation stands out for its ability to fine-tune image synthesis with precision and minimal computational cost. LoRA stands out in its capability for controllable generation, it enables the creation of specific characters, particular types of clothing, unique styles, or other distinctive visual features, and can be trained and later used to produce varied and precise representations of these elements in the generated images. However, existing image generation methodologies utilizing LoRAs encounter limitations in effectively combining multiple LoRAs, particularly as the quantity of LoRAs to be amalgamated increases, thus hindering the composition of complex images. Given this limitation, a critical question emerges: How can we effectively composite multiple trained LoRAs in a training-free manner, while still retaining their unique individual attributes in image generation?

As shown in Figure 2, we find that directly applying pre-trained LoRA modules to compose the image often leads to semantic conflicts. This failure primarily arises because independent LoRAs

---

[*]Mingzhu Shen is the corresponding author.
[1]The source code is released at https://github.com/Yqcca/CMLoRA.

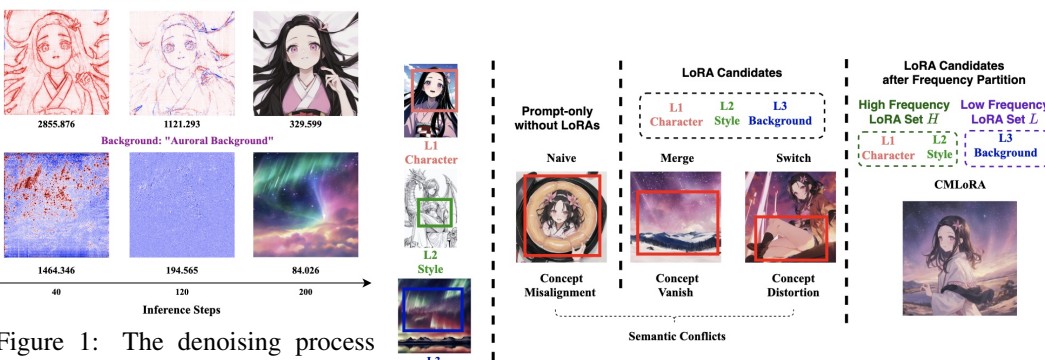

Figure 1: The denoising process with a Character LoRA and a Background LoRA. The plot illustrates the difference in amplitude of high-frequency components $\Delta \mathcal{H}_{0.2}\left(\overline{\mathbf{x}}_t; 40\right)$ between 40-step interval generated by the Character LoRA and Background LoRA after the inverse Fourier Transform, matching each step $t$.

Figure 2: **Observation:** Prompt-only generation (Naive) and existing LoRA combination methods (Merge and Switch) often lead to semantic conflicts. This failure primarily arises because independent LoRAs are integrated to contribute equally to image generation during the denoising process. CMLoRA employs a frequency-domain-based LoRA scheduling mechanism to integrate multiple concept LoRAs, effectively addressing semantic conflicts.

are integrated to contribute equally to image generation during the denoising process. We hypothesize that the difficulty in scaling multiple LoRA modules comes from the "semantic conflicts" among them, as LoRAs are typically trained independently and fuse features with varying amplitudes across different frequency domains to the generated image. When these independent LoRAs are integrated to contribute equally to image generation, inherent conflicts may arise. To investigate LoRA behavior during the denoising process, we shift the perspective to the Fourier domain, a research area that has received limited prior investigation, because of its advantages: 1) efficient image feature detection 2) robustness to noise in the spatial domain (Frank et al., 2020). Figure 1 exhibits a disparity in how Character and Background LoRAs function differently during the denoising process, indicating that their fusion of semantic information with varying amplitudes across different frequency spectra into the generated image. It is evident that the Character LoRA fuses a higher proportion of high-frequency components, resulting in greater variation in edges and textures compared to the Background LoRA, during the inference. This finding suggests that certain LoRAs introduce more pronounced high-frequency modifications during denoising, whereas others primarily influence low-frequency elements. This can be explained as follows: (1) Some LoRAs enhance high-frequency components, corresponding to rapid changes like edges and textures. (2) Others target low-frequency components, representing broader structures and smooth color transitions. Furthermore, high-frequency components are predominantly fused during the early stages of inference, aligning with the observation made in prior work: *high-frequency components vary more significantly than low-frequency ones throughout the denoising process* (Si et al., 2024). Consequently, improper integration of various LoRAs may result in visual artifacts or semantic inconsistencies in the generated images.

In light of the phenomenon found in Figure 1, we propose a Fourier-based method to classify LoRAs with different frequency responses and group them into distinct sets. Through our profiling approach, we categorize LoRAs into high-frequency and low-frequency sets. During inference, high-frequency LoRAs are employed predominantly in the early denoising stages, while low-frequency LoRAs are applied dominantly later. Building upon these insights, we introduce Cached Multi-LoRA (CMLoRA), a novel framework for multi-LoRA composition. During inference, CMLoRA employs a flexible multi-LoRA injection backbone: denoising the noisy image with the predominant contributions of dominant LoRAs, while incorporating supplementary contributions from cached non-dominant LoRAs. Guided by an effective frequency-domain-based scheduling mechanism, CMLoRA selects a dominant LoRA from the high-frequency set in the early stages of denoising, transitioning to a dominant LoRA from the low-frequency set in the later stages. Additionally, we introduce a specialized modulation factor as a hyperparameter, which scales the contribution of the dominant LoRA during inference, providing precise control over its influence on the overall composition. As a result, CMLoRA effectively resolves the semantic conflicts that frequently occur during multi-LoRA composition, enhancing the quality of generated images and computational efficiency.

Our findings, encompassing both qualitative and quantitative results, demonstrate that CMLoRA outperforms existing LoRA composition approaches, and we make the following contributions:

- We introduce a Fourier-based approach for LoRA partition that leverages frequency characteristics. Our method is grounded in frequency-domain profiling of LoRAs during the inference stage of the diffusion model, enabling us to classify LoRAs into high- and low-frequency sets. This classification allows for more effective LoRA fusion by grouping LoRAs with similar frequency behaviors, reducing potential semantic conflicts and improving the coherence of generated images.
- We propose a flexible LoRA composition framework: Cached Multi-LoRA, designed to optimize LoRA integration without requiring additional training. Our method overcomes existing constraints on the number of LoRAs that can be integrated, offering enhanced flexibility, improved image quality, compared to state-of-the-art LoRA integration methods.
- We address the problem of the lack of evaluation methodologies and metrics in multi-LoRA image generation by introducing an improved evaluator built upon MiniCPM-V. We set a new comprehensive benchmark for assessing four aspects of multi-LoRA composition: 1) element integration, 2) spatial consistency, 3) semantic accuracy, and 4) aesthetic quality.

## 2 METHOD

### 2.1 PRELIMINARY

**Text-to-Image Diffusion Models** Diffusion models (Ho et al., 2020; Song et al., 2020) belong to a class of generative models that gradually introduce noise into an image during the forward diffusion process and learn to reverse this process to synthesize images. When combined with pre-trained text embedding, text-to-image diffusion models (Podell et al., 2023; Esser et al., 2024) are capable of generating high-fidelity images based on text prompts. The diffusion model, denoted as $\epsilon_\theta$, with trainable parameters $\theta$, is optimized to predict the noise added to the noisy latent representation $\mathbf{z}_t$, conditioned on the provided text $c$. Typically, a mean-squared error loss function is utilized as the denoising objective:

$$\mathcal{L} = \mathbb{E}_{\mathbf{z},c,\epsilon,t}[||\epsilon - \epsilon_\theta\left(\mathbf{z}_t, t, c\right)||], \tag{1}$$

where $\epsilon \sim \mathcal{N}(0,1)$ is the additive Gaussian noise, $\mathbf{z}_t$ is the latent feature at timestep $t$ and $\epsilon_\theta$ is the denoising U-Net with learnable parameter $\theta$.

**Low-Rank Adaptation** The Low-Rank Adaptation (LoRA) technique freezes the pre-trained model weights and introduces trainable low-rank decomposition matrices into each layer of the neural network architecture, thereby significantly reducing the number of trainable parameters required for downstream tasks (Hu et al., 2021). For a pre-trained weight matrix $W \in \mathbb{R}^{m \times n}$ in a diffusion model $\epsilon_\theta$, we constrain its update by representing the latter with a low-rank decomposition, i.e., updating $W$ to $\hat{W}$, where $\hat{W} = W + \Delta W = W + BA$. $B \in \mathbb{R}^{m \times r}$ and $A \in \mathbb{R}^{r \times n}$ are matrices of a low-rank factor $r$, satisfying $r \ll \min(n, m)$.

### 2.2 LORA DISPARITY BASED ON FOURIER ANALYSIS

We hypothesize that the challenges in scaling multiple LoRA modules stem from the "semantic conflicts" that arise among them, given that LoRAs are typically trained independently and fuse features with varying amplitudes across different frequency domains during the denoising process. Building on the notable disparities in how different LoRAs fuse high-frequency components during the denoising process, as illustrated in Figure 1, we expand our investigation to delineate the specific contributions of various LoRAs within this process and to explore the internal characteristics of LoRAs based on their profiled categories. We aim to establish a Fourier-based method to classify LoRAs according to their frequency responses and group them into distinct sets, as shown in Figure 2. Using our profiling approach, LoRAs are categorized into high-frequency and low-frequency sets. During inference, high-frequency LoRAs are primarily utilized in the early stages of denoising to enhance detail and texture, while low-frequency LoRAs are predominantly applied in the later stages to refine overall structure and coherence.

To evaluate the salient characteristics of the contribution of high-frequency components modification from different LoRAs in the denoising process, we conduct a controlled experiment based on the testbed *CDmposLoRA* (Zhong et al., 2024), comprising of 5 different LoRA categories: Character, Clothing, Style, Background and Object. We first use the diffusion model combined with each LoRA in different profiled LoRA categories to generate a batch of images. Then we use the 2D Fast Fourier Transform (FFT) to transform images from the spatial domain to the frequency domain and

compute their frequency spectrum. Finally, we extract the amplitude of the high-frequency components and compute their change in amplitude during the whole denoising process. Mathematically, these operations are performed as follows:

We first computer the average feature map along the channel dimension by taking the mean:

$$\overline{\mathbf{x}}_t = \frac{1}{C} \sum_{i=1}^{C} \mathbf{x}_{t,i}, \tag{2}$$

where $\mathbf{x}_{t,i}$ represents the $i$-th channel of the feature map $\mathbf{x}_t$ at denoising timestep $t$. $C$ denotes the total number of channels in $\mathbf{x}_t$. Then, we quantify the amplitude of high-frequency components in the generated image by analyzing its distribution across the frequency spectrum. We calculate the 2D FFT for $\overline{\mathbf{x}}_t$ and extract the amplitude of $h$ percentage of high-frequency components in the frequency domain:

$$\mathcal{F}(\overline{\mathbf{x}}_t) = 2\text{DFFT}\left(\overline{\mathbf{x}}_t\right)$$
$$\mathcal{H}_h\left(\overline{\mathbf{x}}_t\right) = \left| \mathcal{F}\left(\overline{\mathbf{x}}_t\right) \cdot \mathbf{1}_{\{\|\mathbf{u}(\mathcal{F}(\overline{\mathbf{x}}_t))\| > h \cdot \max \|\mathbf{u}(\mathcal{F}(\overline{\mathbf{x}}_t))\|\}} \right|, \tag{3}$$

where 2D FFT$(\cdot)$ denotes the 2D Fast Fourier transform averaged in the radial axis, $|\cdot|$ calculates the amplitude of components across the frequency spectrum, and $\mathbf{u}(\cdot)$ calculates the frequency range in the Fourier domain. Subsequently, the change in amplitude of high-frequency components between interval $t-z$ and $t$ is determined as follows:

$$\Delta\mathcal{F}\left(\overline{\mathbf{x}}_t; z\right) = \mathcal{F}\left(\overline{\mathbf{x}}_t\right) - \mathcal{F}\left(\overline{\mathbf{x}}_{t-z}\right)$$
$$\Delta\mathcal{H}_h\left(\overline{\mathbf{x}}_t; z\right) = \left| \Delta\mathcal{F}\left(\overline{\mathbf{x}}_t; z\right) \cdot \mathbf{1}_{\{\|\mathbf{u}(\Delta\mathcal{F}(\overline{\mathbf{x}}_t; z))\| > h \cdot \max \|\mathbf{u}(\Delta\mathcal{F}(\overline{\mathbf{x}}_t; z))\|\}} \right|, \tag{4}$$

Based on Equation (4), we profile the LoRA categories in the testbed through the following steps: 1) Establishing a prioritized LoRA order strategy, denoted as $\mathcal{O}$, by ranking the variation in the intensity of high-frequency components, $\overline{\Delta\mathcal{H}_h\left(\overline{\mathbf{x}}_t; z\right)}$, across different LoRA categories. 2) Using the strategy $\mathcal{O}$, we can categorize LoRAs into a high-frequency dominant set $H$ and a low-frequency dominant set $L$ for a multi-LoRA composition task. 3) LoRAs from the high-frequency dominant set $H$ are employed predominantly during the initial stages of denoising, where their dynamic features can enhance the image's detail and texture. In contrast, LoRAs from the low-frequency dominant set $L$ are utilized primarily in the later stages of the denoising process.

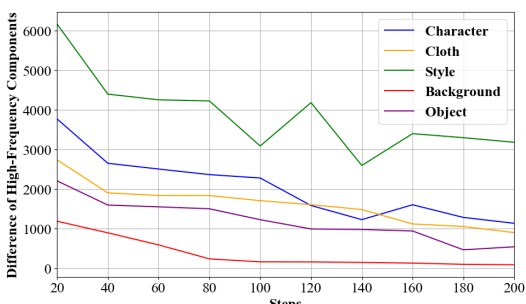

Figure 3: Summary of the change in amplitude of high-frequency components, $\overline{\Delta\mathcal{H}_{0.2}\left(\overline{\mathbf{x}}_t; 20\right)}$, during the denoising process for generated images with LoRAs across different LoRA categories.

We set $h = 0.2$ to conduct our exploration. We calculate the amplitude of high-frequency components, denoted as $\mathcal{H}_{0.2}(\overline{\mathbf{x}}_t)$, and change in amplitude of high-frequency components, represented as $\Delta\mathcal{H}_{0.2}\left(\overline{\mathbf{x}}_t; z\right)$, for certain step $t$ for all LoRAs within the various profiled categories in our testbed. These values are averaged among the LoRAs within their respective categories, for instance, we have multiple LoRAs coming from the Character category or the Cloth category. This then yields $\overline{\mathcal{H}_{0.2}(\overline{\mathbf{x}}_t)}$ and $\overline{\Delta\mathcal{H}_{0.2}\left(\overline{\mathbf{x}}_t; z\right)}$ for all LoRA categories.

We perform the profiling on the following categories: Chracter, Cloth, Style, Background and Object. Figure 3 illustrates the variation in the intensity of high-frequency components, $\overline{\Delta\mathcal{H}_{0.2}\left(\overline{\mathbf{x}}_t; z\right)}$, across all LoRA categories in our testbed throughout the denoising process. Evidently, certain high-frequency dominant LoRAs, such as Style and Character, incorporate larger amplitudes of high-frequency features into the generated image compared to others, particularly during the early stages of inference.

Building upon these insights, we subsequently leverage the change in amplitudes of high-frequency components $\overline{\Delta\mathcal{H}_{0.2}\left(\overline{\mathbf{x}}_t; z\right)}$ as a criterion for selecting LoRA candidates throughout the denoising process. We establish a prioritized LoRA order strategy $\mathcal{O}$ using the ranking of $\overline{\Delta\mathcal{H}_{0.2}\left(\overline{\mathbf{x}}_t; 20\right)}$ across different LoRA categories: Style, Character, Cloth, Object and Background. We aim to maximize the contribution of the LoRAs ranked highest in the order $\mathcal{O}$ during the early stages of the denoising process.

Following the strategy $\mathcal{O}$, we can categorize LoRAs into a high-frequency dominant set $H$ and a low-frequency dominant set $L$ for a multi-LoRA composition task. Specifically, we reserve the LoRA candidate ranked last in the order $\mathcal{O}$ for inclusion in the low-frequency dominant set $L$, while placing the remaining LoRAs into the high-frequency dominant set $H$. As a result, LoRAs from the high-frequency dominant set $H$ are employed predominantly during the initial stages of denoising, where their dynamic features can effectively enhance the image's detail and texture. In contrast, LoRAs from the low-frequency dominant set $L$ are utilized primarily in the later stages of the denoising process. This strategic transition between dominant LoRAs mitigates the semantic conflicts that may arise from the fusion of multiple LoRAs.

## 2.3 CACHED MULTI-LORA

In this paper, we investigate multi-LoRA composition within the context of diffusion models, aligning with prior studies on LoRA merging (Gu et al., 2024; Zhong et al., 2024). Building on the training-free LoRA integration methods introduced by Zhong et al. (2024), we focus on two well-established frameworks: LoRA Switch and LoRA Composite, as outlined in Appendix A.3. Based on our evaluations detailed in Section 3.2, while LoRA Composite injects all activated LoRAs at each timestep during the denoising process, LoRA Switch – activating only one LoRA per timestep – performs better in the multi-LoRA composition task.

Since LoRA Switch outperforms LoRA Composite based on the evaluation in Section 3.2, we hypothesize that LoRA Switch mitigates the "semantic conflicts" in multi-LoRA fusion by limiting activation to a single LoRA per timestep. However, by applying the frequency profiling approach described in Section 2.2 to classify LoRAs into high-frequency ($H$) and low-frequency ($L$) sets, we conjecture that LoRA Composite can potentially surpass LoRA Switch, since the effective LoRA partitioning strategy could allow LoRA Composite to integrate multiple LoRAs efficiently while minimizing semantic conflicts. Thus, we introduce a flexible multi-LoRA framework based on LoRA Composite, termed Cached Multi-LoRA.

### 2.3.1 LORA COMPOSITE

CMLoRA is grounded in the LoRA partition approach illustrated in Section 2.2, focusing on the systematic investigation of optimal dominant LoRA fusion sequences to enhance multi-LoRA integration. Our framework involves calculating both unconditional and conditional score estimates for each LoRA individually at every denoising step. With a set of $N$ LoRAs in place, let $\hat{\theta}_i$ denote the parameters of the diffusion model $e_\theta$ after incorporating the $i$-th LoRA. The collective guidance $\hat{e}(\mathbf{z}_t, c)$ based on textual condition $c$ is derived by aggregating the scores from each LoRA:

$$\hat{e}(\mathbf{z}_t, c) = \frac{1}{N} \sum_{i=1}^{N} w_i[(1-s) \cdot e_{\hat{\theta}_i}(\mathbf{z}_t) + s \cdot e_{\hat{\theta}_i}(\mathbf{z}_t, c)], \tag{5}$$

where $w_i$ is a real scalar weight allocated to each LoRA and $\sum_{i=1}^{N} w_i < \infty$, intended to adjust the influence of the $i$-th LoRA. By aggregating these scores, the technique ensures balanced guidance

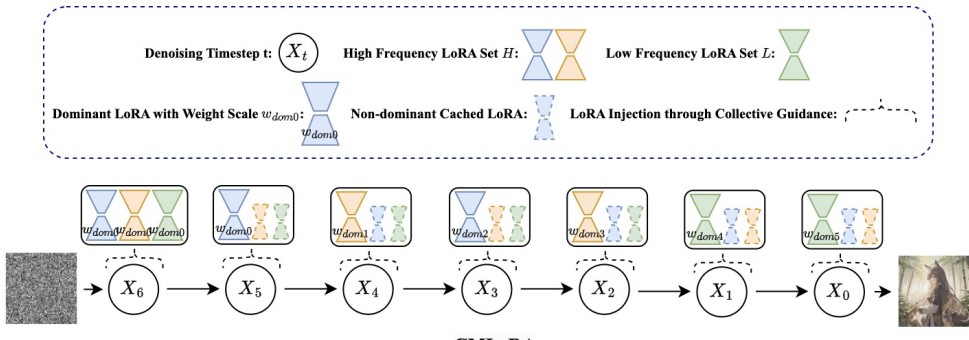

Figure 4: Overview of our multi-LoRA composition framework during a 7-step denoising process. Each color represents a distinct LoRA, where solid shapes indicate dominant LoRAs performing full inference, and hollow shapes represent non-dominant LoRAs leveraging the caching mechanism at their respective steps. The weight scale $w_{dom_i}$ on each dominant LoRA signifies its influence during the denoising process, where $w_{dom_0} = w_{dom_1} > \cdots > w_{dom_5}$.

throughout the image generation process, facilitating the cohesive integration of all elements represented by different LoRAs. Additionally, we introduce a specialized modulation factor $w_{dom}$ as a hyperparameter, which scales the contribution of the dominant LoRA during inference, as demonstrated in **Dominant LoRA Scale** in Appendix F.

Figure 4 illustrates the main features of our multi-LoRA composition framework:

- Dominant LoRA swaps among LoRAs in the high-frequency dominant set $H$ per timestep.
- The weight scale of dominant LoRA, $w_{dom} \in \mathbb{R}$, is decaying during the denoising process.
- Non-dominant LoRAs use the caching mechanism demonstrated in Section 2.3.2.

### 2.3.2 THE CACHING MECHANISM

To amplify the contribution of the determined dominant LoRA and ensure more stable frequency fusion in multi-LoRA composition, we further introduce caching strategies for non-dominant LoRAs during the denoising process.

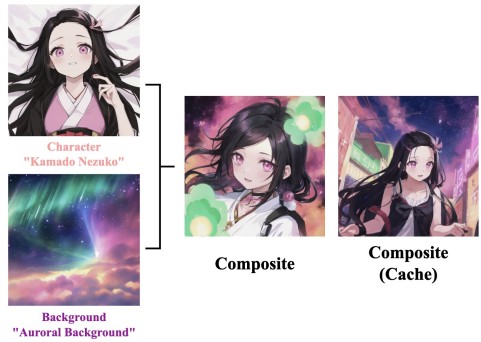

Figure 5: Character LoRA and Background LoRA composition. Visual artifacts (green flowers) appear in the image generated by LoRA Composite framework, as illustrated in Appendix A.3. Introducing the caching mechanism can alleviate the semantic conflict we have here.

Figure 6: The percentage of steps with a similarity greater than 0.9 to the current step for cached latent feature maps, which is the output of the up-sampling block $U_2^t$ of U-net with one LoRA during LoRA composition.

Figure 5 illustrates the semantic conflicts that arise in LoRA Composite framework, as discussed in Appendix A.3, although we have performed a partitioning already in the frequency domain. Since LoRA Composite assigns equal weights to all frequency components of the LoRAs during the denoising process, it is prone to feature conflicts in the fused outputs of different LoRAs in Fourier space. Drawing inspiration from the DeepCache technique proposed by Ma et al. (2024), we investigate how the caching mechanism can be leveraged as a potential remedy.

According to the reverse process, $\mathbf{z}_{t-1}$ is conditionally generated based on the previous result $\mathbf{z}_t$. Initially, we generate $\mathbf{z}_t$ in the same way as usual, where the calculations are performed across all entire U-Nets $\{e_{\hat{\theta}_1}, \cdots, e_{\hat{\theta}_N}\}$ incorporating LoRAs with weight matrices $\{W_1, \cdots, W_N\}$. To obtain the next output $\mathbf{z}_{t-1}$, we retrieve the high-level features produced in the previous collective guidance $\hat{e}(\mathbf{z}_t, c)$. Specifically, consider a skip branch $s^i$ in the U-Net incorporating the $i$-th LoRA $e_{\hat{\theta}_i}$, which bridges $D_{s^i}$ and $U_{s^i}$, we cache the feature maps from the previous up-sampling block at the time $t$ as the following:

$$F_{i,\text{cache}}^t \leftarrow U_{i,m+1}^t(\cdot), \tag{6}$$

which is the feature from the main branch at timestep $t$. These cached features are reused in subsequent inference steps. At the next timestep, $t-1$, if full inference for the U-Net incorporating the $i$-th LoRA $e_{\hat{\theta}_i}$ is unnecessary, we perform a dynamic partial inference. Based on the previously generated collective guidance $\hat{e}(\mathbf{z}_t, c)$, we compute only the necessary components for the $m$-th skip branch, while substituting the main branch computation with a retrieval operation from the cache in Equation 6. Thus, for the U-Net incorporating the $i$-th LoRA, the input for $U_{i,m}^t$ at timestep $t-1$ is formulated as:

$$\text{Concat}(D_{i,m}^{t-1}(\cdot), F_{i,\text{cache}}^t), \tag{7}$$

where $D_{i,m}^{t-1}(\cdot)$ represents the output of the $m$-th down-sampling block.

**Cache Interval Determination**    As shown in Figure 6, the percentage of steps with a similarity greater than 0.9 between cached latent feature maps and the current step follows a distinct trend across all LoRAs in LoRA Composite framework. To capture the similarity trend of feature maps fused by LoRAs, we propose a non-uniform caching interval strategy with two specialized hyper-parameters: $c_1, c_2 \in \mathbb{Z}$. These hyper-parameters control the strength of the caching behavior during inference. Specifically, for a denoising process with $T$ timesteps, the sequence of timesteps that performs full inference is:

$$
\begin{aligned}
\mathcal{I} &= \mathcal{I}_1 \cup \mathcal{I}_2 \cup \mathcal{I}_3 \\
\mathcal{I}_1 &= \{c_1 \cdot t \mid 0 \leq c_1 \cdot t < \lfloor 0.4 \cdot T \rfloor, \text{ where } t \in \mathbb{Z}\} \\
\mathcal{I}_2 &= \{\lfloor 0.4 \cdot T \rfloor + c_2 \cdot t \mid \lfloor 0.4 \cdot T \rfloor \leq c_2 \cdot t < \lfloor 0.9 \cdot T \rfloor, \text{ where } t \in \mathbb{Z}\} \\
\mathcal{I}_3 &= \{\lfloor 0.9 \cdot T \rfloor + c_1 \cdot t \mid \lfloor 0.9 \cdot T \rfloor \leq c_1 \cdot t < T, \text{ where } t \in \mathbb{Z}\}.
\end{aligned}
\tag{8}
$$

The interval $[\lfloor 0.4 \cdot T \rfloor, \lfloor 0.9 \cdot T \rfloor]$ are established based on the condition that the similarity of the cached features exceeds 20% with a 90% confidence interval, as demonstrated in **Caching Interval and Modulation Hyper-parameters** in Appendix F. This strategic approach aims to further mitigate the issue of semantic conflict in multi-LoRA composition. Notably, it offers two key advantages. First, it amplifies the features contributed by the dominant LoRA feature map, while minimizing changes in features fused from non-dominant LoRAs, allowing the dominant LoRA to play a more critical role during the denoising process. Second, it mitigates the negative effects of frequency conflicts in the Fourier domain, thereby achieving a more balanced trade-off between multi-concept fusion and texture preservation for the activated LoRAs during the inference. The effectiveness of our proposed caching strategy is discussed in Section 3.2.2.

In summary, by leveraging a Fourier-based approach to partition LoRAs based on their frequency characteristics, we can determine the optimal order of dominant LoRA application during the denoising process. Building on this partitioning strategy, we introduce CMLoRA, a novel LoRA composition framework that employs a flexible multi-LoRA injection backbone: denoising the noisy image with the predominant contributions of dominant LoRAs, while incorporating supplementary contributions from cached non-dominant LoRAs.

## 3 EXPERIMENTS

### 3.1 EXPERIMENTAL SETUP

**Models and Evaluation Setup**    We begin by examining the prompt-only generative method without incorporating concept LoRAs (Rombach et al., 2022), denoted as the naive model. In addition, we utilize training-free multi-LoRA composition methods with the same text prompt used in the naive model, including LoRA Switch (Zhong et al., 2024), LoRA Composite (Zhong et al., 2024), and LoRA Merge (HuggingFace, 2023). Alongside these methods, we incorporate the LoraHub (Huang et al., 2023) framework, which fluidly combines multiple LoRA modules with few-shot learning, as our baseline. For each baseline method, we apply our proposed caching method, denoting the results as LoRA Switch (Cache$D$), LoRA Composite (Cache$D$), and LoRA Merge (Cache$_D$), respectively. We also consider a uniform cache interval strategy governed by a hyper-parameter $c$. For all $T$ denoising steps, the sequence of timesteps that performs full inference is defined as:

$$
\mathcal{I} = \{c \cdot t \mid 0 \leq c \cdot t \leq T, \text{where } t \in \mathbb{Z}\}.
\tag{9}
$$

This extension allows us to integrate multi-LoRA composition methods with varying uniform caching strategies for evaluation, as analyzed in Appendix F.2. Additionally, we integrate our proposed LoRA partitioning strategy to LoRA Switch also as a baseline, referred as Switch-A. The analysis of computational cost is detailed in Appendix B.2.

Based on the testbed *ComposLoRA* (Zhong et al., 2024), we curate two unique subsets of LoRAs representing realistic and anime styles. Each subset comprises a variety of elements: 3 characters, 2 types of clothing, 2 styles, 2 backgrounds, and 2 objects, culminating in a total of 22 LoRAs. We discuss the challenges in constructing a well-defined multi-LoRA composition testbed in Appendix D.

**Evaluation Metrics**    We employ CLIPScore (Hessel et al., 2022) and ImageReward (Xu et al., 2023) to evaluate the comprehensive image generation capabilities of all multi-LoRA composition

methods. While both metrics perform effectively within their evaluation domains, we find that they struggle to accurately assess out-of-distribution (OOD) concepts. Notably, ImageReward exhibits more pronounced issues with OOD instances compared to CLIPScore, as illustrated in Table 4 (which assigns negative scores to images generated by anime LoRAs). We discuss this limitation in Appendix D. Consequently, we include CLIPScore as a key evaluation metric to assess the efficacy of multi-LoRA composition frameworks at first.

Recent advancements in multi-modal large language models (LLMs), such as Yao et al. (2024); OpenAI (2024); Yang et al. (2023), have shown significant potential in multi-modal tasks, positioning them as promising tools for evaluating image generation. In this study, we harness capabilities of MiniCPM-V (Yao et al., 2024), an end-side multi-modal LLM designed for vision-language understanding, to evaluate composable image generation, utilizing in-context few-shot learning to address challenges posed by OOD concepts. The full evaluation process is provided in Appendix E.

**Implementation Details**   We use the open-source platform Diffusers (von Platen et al., 2022) as the pipeline to conduct our experiments. We employ stable-diffusion-v1.5 implemented by Rombach et al. (2022) in PyTorch (Paszke et al., 2019) as the backbone model. For the anime style subset, the settings differ slightly with 200 denoising steps, a guidance scale $s$ of 10, and an image size of $512 \times 512$. The DPM-Solver++ proposed by Lu et al. (2022) is used as the scheduler in the generation process. The LoRA scale for all LoRAs is set to $1.4$, which is applied within the cross-attention module of the U-Net. The dominant weight scale $w_{\text{dom}}$ is initially set at $N - 0.5$, where $N$ is the total number of activated LoRAs. Then this weight scale is adjusted using a decaying method. For the $i$-th turn of switching the dominant LoRA, the weight is defined as: $w_{\text{dom}}^{i} = w_{\text{dom}}^{i-1} - 0.5^{i}$. In addition, we select $c_1 = 2$ and $c_2 = 3$ for the caching strategy applied to non-dominant LoRAs. The hyper-parameters are selected using grid search methods described in Appendix F.

## 3.2   RESULTS

### 3.2.1   CLIPSCORE EVALUATION

Table 1: ClipScore for Selected multi-LoRA Composition Methods.

| Model | ClipScore | | | |
|---|---|---|---|---|
| | N=2 | N=3 | N=4 | N=5 |
| Naive (Rombach et al., 2022) | 35.014 | 34.927 | 34.384 | 33.809 |
| Merge (HuggingFace, 2023) | 33.726 | 34.139 | 33.399 | 32.364 |
| Switch (Zhong et al., 2024) | 35.394 | 35.107 | 34.478 | 33.475 |
| Composite (Zhong et al., 2024) | 35.073 | 34.082 | 34.802 | 32.582 |
| LoraHub (Huang et al., 2023) | **35.681** | 35.127 | 34.970 | 33.485 |
| Switch-A | 35.451 | **35.383** | 34.877 | 33.366 |
| Merge (Cache$_D$) | 33.554 | 33.917 | 33.465 | 32.654 |
| Switch (Cache$_D$) | 35.528 | 35.112 | 34.845 | 34.056 |
| Composite (Cache$_D$) | 35.295 | 34.984 | 34.981 | 33.097 |
| LoraHub (Cache$_D$) | 35.609 | 34.919 | 35.135 | 33.659 |
| Switch-A (Cache$_D$) | 35.139 | **35.383** | 34.930 | 34.250 |
| CMLoRA (Cache$_D$) | 35.422 | 35.215 | **35.208** | **34.341** |

We first present the comparative evaluation results obtained using CLIPScore (Hessel et al., 2022). Table 1 presents the ClipScore performance for several LoRA composition methods across different numbers of LoRAs ($N = 2$ to $N = 5$). Additional experimental results are included in Appendix B and visualization demonstrations are provided in Appendix C. Three key observations emerge from the analysis:

- *In general, LoRA Switch shows higher CLIPScore than LoRA Composite across most cases.* For $N = 5$, LoRA Switch scores 33.475, outperforming LoRA Composite's 32.582. This trend indicates that LoRA Switch handles multi-LoRA integration more effectively than LoRA Composite, particularly when the number of LoRAs increases. However, Switch-A (LoRA Switch with frequency partitioning), does not offer a better performance compared to CMLoRA with large $N$ values, which confirms our hypothesis in Section 2.2.
- *Our proposed method, CMLoRA with dynamic caching (Cache$_D$), consistently delivers the highest or near-highest CLIPScore across all scenarios.* For $N = 3$, CMLoRA achieves a CLIPScore of 35.215, outperforming both LoraHub (34.919) and Switch-A (35.383). Similarly, for $N = 5$, CMLoRA records the highest score of 34.341, outperforming other methods. These results highlight the efficiency and robustness of CMLoRA in the multi-LoRA integration task.

- *The task of multi-concept image generation remains highly challenging, especially as the number of elements to be composed increases.* As the number of LoRAs increases from $N = 2$ to $N = 5$, the CLIPScore of generated images generally decreases across all methods. This trend highlights the increasing challenge of compositional image generation when more elements are involved.

*CLIPScore's evaluations fall short in assessing specific compositional and quality aspects due to its inability to discern the nuanced features of each element* (Zhong et al., 2024). To address this limitation and provide a more thorough analysis, we complement our findings with an MLLM evaluation across all multi-LoRA composition methods.

### 3.2.2 MiniCPM-V-based Evaluation

The evaluation using MiniCPM-V involves scoring the performance of CMLoRA (Cache$_D$) versus others across four dimensions, as well as determining the win/loss rate based on these scores. The specific score and win/loss rate are illustrated in the radar map Figure 7 and the win/loss rate plot: Figure 8. Additional experimental results are available in Appendix B and visualization demonstrations are presented in Appendix C. As demonstrated in Table 6, the Naive model exhibits the lowest score in Semantic Accuracy, highlighting that incorporating multiple LoRA mechanisms can significantly enhance the generative model's performance in multi-concept image generation. Based on our observations, we conclude that our proposed CMLoRA achieves significant improvements across various metrics for multi-concept image generation, particularly in aesthetic quality. Moreover, CMLoRA demonstrates superior overall composition quality, with a win rate that is $20\%$ higher than LoRA Merge and $10\%$ higher than other methods.

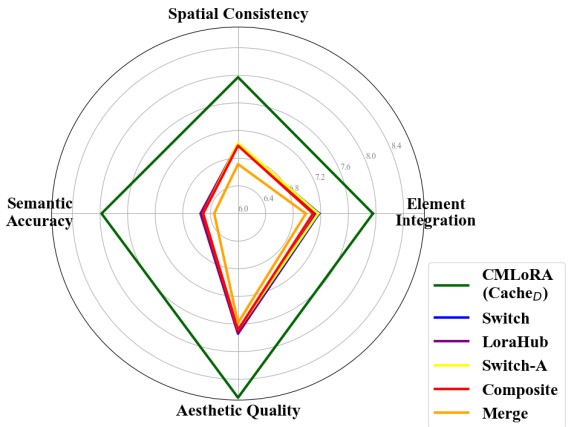

Figure 7: The performance evaluation results of LoRA integration methods on the ComposLoRA testbed using MiniCPM-V are presented. Detailed scores are available in Table 6.

Figure 8: Comparison of CMLoRA (Cache$_D$) against other Multi-LoRA composition methods based on win rate.

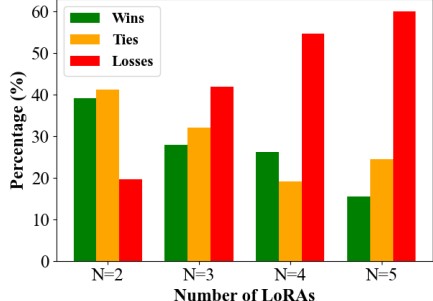

Figure 9: Comparison of Composite against Composite(Cache$_D$) based on win rate.

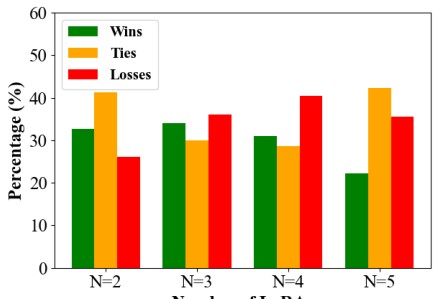

Figure 10: Comparison of CMLoRA against CMLoRA(Cache$_D$) based on win rate.

In addition, to verify our hypothesis presented in Section 2.3.2 – caching non-dominant LoRAs strategically during the denoising process can amplify the contribution of the determined dominant LoRA and ensure more stable frequency fusion in multi-LoRA composition – we assess both LoRA Composite and CMLoRA, including their caching versions, based on win rate. Figure 9 and Figure 10 illustrate that the caching strategy can significantly improve the quality of multi-concept

image generation. Specifically, LoRA Composite benefits more from the caching strategy compared to CMLoRA because LoRA Composite does not implement the LoRA partition strategy during the denoising process. In summary, we conclude the following results:

- *CMLoRA demonstrates greater potential than other multi-LoRA fusion methods in addressing the semantic conflicts that arise during multi-LoRA composition.* The MiniCPM-V evaluation further reinforces CMLoRA's advantages, particularly in aesthetic quality, where it achieves a higher win rate than other competing methods in the multi-LoRA composition image generation task.
- *Our analysis confirms that strategically caching non-dominant LoRAs enhances the performance of dominant LoRAs during the denoising process.* Notably, LoRA Composite benefits more from this caching strategy compared to CMLoRA, likely due to the absence of the Fourier LoRA partition approach during denoising in LoRA Composite.

## 4 RELATED WORK

### 4.1 MULTI-CONCEPT TEXT-TO-IMAGE GENERATION

Multi-concept composable image generation plays a crucial role in digital content customization, allowing for the creation of images that align with predefined specifications. Existing research in this domain primarily focuses on the following approaches: adjusting the generative processes of diffusion models to better align with specified requirements (Jiang et al., 2024; Kumari et al., 2023; Xiao et al., 2024; Lin et al., 2024), or integrating a series of independent modules that impose desired constraints (Kwon et al., 2024; Gu et al., 2024; Zhong et al., 2024).

While traditional methods excel at producing images based on general concepts, they often struggle with the precise integration of user-defined objects (Kumari et al., 2023; Gafni et al., 2022). Other approaches can compose specific objects into images but often require extensive fine-tuning and struggle with multiple objects simultaneously (Huang et al., 2023; Shah et al., 2023). To address these limitations, we propose a training-free, instance-level LoRA composition framework, which enables the accurate assembly of user-specified elements in image generation.

### 4.2 MULTI-LORA INTEGRATION MANIPULATIONS

Recent research has focused on leveraging large language models (LLMs) or diffusion models as base models, aiming to manipulate LoRA weights for various objectives. These include element composition in image generation (Yang et al., 2024; Shah et al., 2023), reducing the parameters needed for multi-modal inference (Chen et al., 2024; Chavan et al., 2023), and adapting models for domain-specific applications (Zhang et al., 2023; Kong et al., 2024; Li et al., 2024). In the realm of LoRA composition techniques, approaches like LoraHub (Huang et al., 2023) utilize few-shot demonstrations to learn coefficient matrices for merging LoRAs, allowing for the fusion of multiple LoRAs into a single new LoRA. LoRA Merge (HuggingFace, 2023) employs addition and negation operators to merge LoRA weights through arithmetic operations. Different from weight-based composition methods, LoRA Switch and LoRA Composite (Zhong et al., 2024) maintain all LoRA weights intact and manipulate the interactions between LoRAs during inference.

Nevertheless, these methods often lead to instability in the merging process as the number of LoRAs increases, leading to semantic conflicts and visual artifacts of generated images. Additionally, they do not adequately utilize the interactive dynamics between the LoRA models and the base model. To address these challenges, our study proposes a novel perspective: analyzing the contributions of different LoRAs in the Fourier domain and developing a novel LoRA-composition framework to mitigate the semantic conflicts arising from multi-LoRA composition.

## 5 CONCLUSION

In this study, we investigate the contributions of various LoRAs to multi-LoRA composition in the Fourier domain. To understand the origins of semantic conflicts in multi-LoRA image generation, we perform a frequency-based analysis of their latent feature maps, revealing significant disparities in frequency contributions among different LoRAs. This finding leads us to propose a versatile Fourier-based profiling method to sequence the fusion of dominant LoRAs during inference, which can be seamlessly integrated into various multi-LoRA frameworks. Finally, we introduce a powerful training-free framework called Cached Multi-LoRA, which denoises images by prioritizing the contributions of dominant LoRAs while incorporating supplementary information from cached non-dominant LoRAs. To validate our approach, we evaluate CMLoRA on the *ComposLoRA* testbed and introduce a scalable, automated evaluation framework based on MiniCPM-V, designed to avoid out-of-distribution issues.

ACKNOWLEDGMENTS

We thank our anonymous reviewers for their constructive comments and insightful feedback.

This work was performed using the Sulis Tier 2 HPC platform hosted by the Scientific Computing Research Technology Platform at the University of Warwick. Sulis is funded by EPSRC Grant EP/T022108/1 and the HPC Midlands+ consortium. Mingzhu Shen is funded by Imperial President's PhD Scholarships. For the purpose of open access, the authors have applied a Creative Commons Attribution (CC BY) licence to any Author Accepted Manuscript version arising.

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

# A LoRA-BASED MANIPULATIONS

## A.1 LoRA MERGE THROUGH A WEIGHT FUSION PERSPECTIVE

**Component-wise Composition of LoRA** LoRA Merge is usually realized by linearly combining multiple LoRAs to synthesize a unified LoRA, subsequently plugged into the diffusion model. Formally, when introducing $N$ distinct LoRAs, the consequent updated matrix $\hat{W}$ in $\epsilon_\theta$ is given by:

$$\hat{W} = W + \sum_{i=1}^{N} w_i \Delta W_i = W + \sum_{i=1}^{N} w_i B_i A_i, \tag{10}$$

where $\sum_{i=1}^{N} w_i = 1$ (HuggingFace, 2023). This manner prevents any adverse impact on the embedding of the original model, but it leads to the loss of individual LoRA characteristics, as the composition weight $w_i$ for each trained LoRA is reduced.

**Element-wise Composition of LoRA** This process integrates the corresponding parameters of the LoRA modules, requiring the modules being combined to have the same rank $r$ to properly align the structures (Huang et al., 2023). Given that $\Delta W_i = B_i A_i$, the combined LoRA module $\hat{W}$ can be obtained by:

$$\hat{W} = (w_1 B_1 + w_2 B_2 + \cdots + w_N B_N)(w_1 A_1 + w_2 A_2 + \cdots + w_N A_N), \tag{11}$$

where the set of optimal weights $\{w_1, w_2, \cdots, w_N\}$ are trained through a black-box optimization.

## A.2 LoRA MERGE THROUGH A GRADIENT FUSION PERSPECTIVE

Compared to weight fusion, gradient fusion aligns the inference behavior of each individual concept, significantly reducing identity loss (Gu et al., 2024). The gradient fusion method first decodes the individual concepts using their respective LoRA weights. It then extracts the input and output features associated with each LoRA layer. These input/output features from different concepts are concatenated, and fused gradients are used to update each layer $W$ using the following objective:

$$W = \arg\min_{W} \sum_{i=1}^{N} ||(W_0 + \Delta W_i)X_i - WX_i||_F^2, \tag{12}$$

where $X_i$ represents the input activation of the $i$-th concept and $|| \cdot ||$ denotes the Frobenius norm.

## A.3 LoRA MERGE THROUGH A DECODING-CENTRIC PERSPECTIVE

**LoRA Switch** With a set of $N$ LoRAs, the methodology initiates with a prearranged sequence of permutations. Starting from the first LoRA, the model transitions to the subsequent LoRA every $\tau$ step (Zhong et al., 2024). The active LoRA at each denoising timestep $t$, ranging from 1 to the total number of steps required, is determined by the following equations:

$$\lambda = \lfloor ((t-1)\text{mod}(N\tau))/\tau \rfloor + 1,$$
$$\hat{W}_t = W + w_i \Delta W_i = W + w_i B_i A_i, \tag{13}$$

where $i$ indicates the index of the currently active LoRA, iterating from 1 to $N$, $\lfloor \cdot \rfloor$ is the floor function, and the weight matrix $\hat{W}_t$ is updated to reflect the contribution from the weighted active LoRA $w_i \Delta W_i$. *By selectively enabling one LoRA at a time, LoRA Switch ensures focused attention to the details pertinent to the current element, thus preserving the integrity and quality of the generated image throughout the process* (Zhong et al., 2024).

**LoRA Composite** LoRA Composite method involves calculating both unconditional and conditional score estimates for each LoRA individually at every denoising step. *By aggregating these scores, the technique ensures balanced guidance throughout the image generation process* (Zhong et al., 2024).

With $N$ LoRAs in place, let $\hat{\theta}_i$ denote the parameters of the diffusion model $e_\theta$ after incorporating the $i$-th LoRA. The collective guidance $\hat{e}(z_t, c)$ based on textual condition $c$ is derived by aggregating the scores from each LoRA, as depicted in the equation below:

$$\hat{e}(z_t, c) = \frac{1}{N} \sum_{i=1}^{N} w_i[(1 - s) \cdot e_{\hat{\theta}_i}(z_t) + s \cdot e_{\hat{\theta}_i}(z_t, c)], \tag{14}$$

where $w_i$ is a scalar weight allocated to each LoRA, intended to adjust the influence of the $i$-th LoRA (Zhong et al., 2024).

## A.4 CMLoRA

### A.4.1 Relation to Classifier Free Guidance

With a set of $N$ LoRAs in place, let $\hat{\theta}_i$ denote the parameters of the diffusion model $e_\theta$ after incorporating the $i$-th LoRA. For a generative model $e_\theta$ integrated with $i$-th LoRA, its classifier-free guidance $\tilde{e}_{\hat{\theta}_i}(\mathbf{z}_t, c)$ based on textual condition $c$ is:

$$(1 - s) \cdot e_{\hat{\theta}_i}(\mathbf{z}_t) + s \cdot e_{\hat{\theta}_i}(\mathbf{z}_t, c). \tag{15}$$

The collective guidance $\hat{e}(\mathbf{z}_t, c)$ based on textual condition $c$ is derived by aggregating the scores from the generative model integrated with each LoRA:

$$\hat{e}(\mathbf{z}_t, c) = \frac{1}{N} \sum_{i=1}^{N} w_i[(1 - s) \cdot e_{\hat{\theta}_i}(\mathbf{z}_t) + s \cdot e_{\hat{\theta}_i}(\mathbf{z}_t, c)], \tag{16}$$

where $w_i$ is a real scalar weight allocated to each LoRA and $\sum_{i=1}^{N} w_i < \infty$, intended to adjust the influence of the $i$-th LoRA.

By aggregating these scores, the technique ensures harmonized guidance throughout the image generation process, facilitating the cohesive integration of all elements represented by different LoRAs.

# B EXPERIMENTAL RESULTS

## B.1 IMAGEREWARD AND CLIPSCORE

To ensure the reliability of our experimental results, we conduct image generation using three random seeds. All reported results in this paper represent the average evaluation scores across these three runs. The experiments were run with a mix of NVIDIA A100 GPUs with 40GB memory and NVIDIA V100 GPUs with 16GB memory. The total amount of inference time for all multi-LoRA composition methods under all metrics is around 1300 GPU hours.

Table 2: Comparison of ImageReward and ClipScore with the selected LoRA integration methods under different number of Anime LoRAs in the *ComposLoRA* testbed.

| Model | ImageReward | | | | ClipScore | | | |
|---|---|---|---|---|---|---|---|---|
| | N=2 | N=3 | N=4 | N=5 | N=2 | N=3 | N=4 | N=5 |
| Merge (Cache$_D$) | 0.359 | 0.227 | -0.462 | -0.506 | 34.974 | 35.147 | 34.143 | 32.705 |
| Merge | 0.357 | 0.211 | -0.346 | -0.590 | 35.136 | 35.421 | 34.164 | 32.636 |
| Merge (Cache$_{c=2}$) | 0.353 | 0.187 | -0.428 | -0.662 | 34.930 | 35.139 | 33.584 | 32.137 |
| Merge (Cache$_{c=3}$) | 0.320 | 0.165 | -0.444 | -0.674 | 34.520 | 34.591 | 34.063 | 31.560 |
| Merge (Cache$_{c=5}$) | 0.313 | 0.006 | -0.484 | -0.586 | 34.653 | 34.669 | 33.450 | 31.924 |
| Switch (Cache$_D$) | 0.416 | 0.201 | -0.176 | -0.322 | 35.510 | 35.576 | 35.367 | 35.347 |
| Switch | 0.424 | 0.207 | -0.184 | -0.306 | 35.285 | 35.482 | 34.532 | 34.148 |
| Switch (Cache$_{c=2}$) | 0.405 | 0.184 | -0.182 | -0.494 | 35.141 | 34.814 | 34.335 | 34.113 |
| Switch (Cache$_{c=3}$) | 0.389 | 0.157 | -0.186 | -0.464 | 35.175 | 35.468 | 34.285 | 34.113 |
| Switch (Cache$_{c=5}$) | 0.378 | 0.105 | -0.186 | -0.466 | 35.130 | 35.189 | 34.151 | 33.799 |
| Composite(Cache$_D$) | 0.366 | 0.125 | -0.226 | **-0.288** | 35.120 | 35.131 | 34.589 | 33.888 |
| Composite | 0.360 | 0.147 | -0.214 | -0.588 | 34.343 | 34.378 | 34.161 | 32.936 |
| Composite (Cache$_{c=2}$) | 0.349 | 0.144 | -0.268 | -0.612 | 33.694 | 34.286 | 33.747 | 32.937 |
| Composite (Cache$_{c=3}$) | 0.333 | 0.136 | -0.278 | -0.780 | 33.719 | 33.950 | 33.817 | 32.855 |
| Composite (Cache$_{c=5}$) | 0.329 | 0.084 | -0.298 | -0.810 | 33.661 | 33.731 | 33.650 | 32.422 |
| LoraHub(Cache$_D$) | 0.388 | 0.157 | -0.410 | -0.574 | 35.257 | 35.221 | 34.582 | 34.412 |
| LoraHub | 0.399 | 0.154 | -0.340 | -0.532 | 35.316 | 35.525 | 34.476 | 33.885 |
| LoraHub (Cache$_{c=2}$) | 0.364 | 0.040 | -0.408 | -0.624 | 35.094 | 35.164 | 34.291 | 32.049 |
| LoraHub (Cache$_{c=3}$) | 0.366 | 0.032 | -0.476 | -0.652 | 35.172 | 35.100 | 33.538 | 31.764 |
| LoraHub (Cache$_{c=5}$) | 0.330 | 0.027 | -0.496 | -0.666 | 34.834 | 34.928 | 33.662 | 30.694 |
| CMLoRA (Cache$_D$) | 0.421 | **0.244** | -0.160 | -0.324 | 35.543 | 35.600 | 35.528 | 35.355 |
| CMLoRA | 0.455 | 0.242 | **-0.112** | -0.310 | 35.556 | 35.555 | **35.791** | **35.691** |
| Switch-A (Cache$_D$) | 0.346 | 0.170 | -0.258 | -0.415 | 35.484 | **35.964** | 35.265 | 35.655 |
| Switch-A | **0.525** | 0.227 | -0.205 | -0.488 | **35.705** | 35.912 | 35.661 | 34.479 |
| Switch-A (Cache$_{c=2}$) | 0.278 | 0.226 | -0.241 | -0.419 | 34.465 | 35.589 | 35.296 | 35.404 |
| Switch-A (Cache$_{c=3}$) | 0.386 | -1.602 | -0.267 | -0.435 | 35.301 | 35.089 | 35.450 | 35.430 |
| Switch-A (Cache$_{c=5}$) | 0.484 | 0.109 | -0.226 | -0.493 | 35.036 | 35.371 | 35.511 | 35.065 |
| CMLoRA (Cache$_{c=2}$) | 0.417 | 0.214 | -0.174 | -0.352 | 35.222 | 35.548 | 35.432 | 35.063 |
| CMLoRA (Cache$_{c=3}$) | 0.382 | 0.179 | -0.190 | -0.406 | 34.876 | 35.076 | 35.359 | 34.823 |
| CMLoRA (Cache$_{c=5}$) | 0.389 | 0.196 | -0.262 | -0.402 | 34.440 | 35.426 | 34.742 | 34.222 |

Table 3: Comparison of ImageReward and ClipScore with the selected LoRA integration methods under different number of Reality LoRAs in the *ComposLoRA* testbed.

| Model | ImageReward | | | | ClipScore | | | |
|---|---|---|---|---|---|---|---|---|
| | N=2 | N=3 | N=4 | N=5 | N=2 | N=3 | N=4 | N=5 |
| Merge (Cache$_D$) | 0.849 | 0.717 | 0.071 | -0.274 | 32.133 | 32.686 | 32.787 | 32.603 |
| Merge | 0.922 | 0.741 | 0.145 | -0.299 | 32.316 | 32.857 | 32.633 | 32.091 |
| Merge (Cache$_{c=2}$) | 0.805 | 0.726 | 0.015 | -0.376 | 31.925 | 32.759 | 32.408 | 31.534 |
| Merge (Cache$_{c=3}$) | 0.796 | 0.663 | -0.011 | -0.508 | 31.253 | 32.600 | 32.588 | 32.449 |
| Merge (Cache$_{c=5}$) | 0.771 | 0.572 | 0.007 | -0.445 | 32.067 | 32.112 | 32.408 | 32.017 |
| Switch (Cache$_D$) | 1.225 | **1.233** | 0.952 | 0.642 | 35.545 | 34.647 | 34.323 | 32.765 |
| Switch | 1.217 | 1.141 | 0.864 | 0.554 | 35.502 | 34.731 | 34.424 | 32.801 |
| Switch (Cache$_{c=2}$) | 0.839 | 1.116 | 0.835 | 0.405 | 34.636 | 34.305 | 35.050 | 33.373 |
| Switch (Cache$_{c=3}$) | 1.051 | 1.047 | 0.812 | 0.578 | 35.189 | 34.469 | 35.056 | 33.372 |
| Switch (Cache$_{c=5}$) | 1.038 | 1.037 | 0.725 | 0.491 | 35.296 | 34.316 | 34.376 | 33.203 |
| Composite (Cache$_D$) | 1.224 | 0.934 | 0.667 | 0.498 | 35.469 | 33.836 | 35.373 | 32.305 |
| Composite | 1.011 | 0.933 | 0.654 | 0.380 | 35.804 | 33.786 | 35.443 | 32.228 |
| Composite (Cache$_{c=2}$) | 1.040 | 0.956 | 0.638 | 0.227 | 35.469 | 34.393 | 35.309 | 31.921 |
| Composite (Cache$_{c=3}$) | 0.992 | 0.889 | 0.628 | 0.158 | 35.070 | 33.857 | 35.373 | 31.852 |
| Composite (Cache$_{c=5}$) | 0.881 | 0.870 | 0.617 | 0.176 | 34.864 | 33.623 | 34.297 | 31.348 |
| LoraHub (Cache$_D$) | 1.093 | 1.077 | 0.923 | 0.680 | 35.961 | 34.617 | **35.687** | 32.905 |
| LoraHub | 1.096 | 1.127 | 0.907 | 0.693 | **36.045** | 34.729 | 35.463 | 33.084 |
| LoraHub (Cache$_{c=2}$) | 1.033 | 1.025 | 0.905 | 0.631 | 35.486 | 34.470 | 34.687 | 32.752 |
| LoraHub (Cache$_{c=3}$) | 1.059 | 0.992 | 0.873 | 0.642 | 35.277 | 34.355 | 34.561 | 32.727 |
| LoraHub (Cache$_{c=5}$) | 0.907 | 0.991 | 0.885 | 0.651 | 35.271 | 34.254 | 34.510 | 32.590 |
| CMLoRA (Cache$_D$) | 1.216 | 1.156 | **1.051** | **0.697** | 35.302 | 34.829 | 34.888 | 33.326 |
| CMLoRA | 1.165 | 1.188 | 0.942 | 0.613 | 35.559 | **35.842** | 34.501 | **33.588** |
| Switch-A (Cache$_D$) | **1.262** | 1.094 | 0.758 | 0.621 | 34.793 | 34.802 | 34.595 | 32.845 |
| Switch-A | 0.817 | 1.096 | 0.811 | 0.591 | 35.196 | 34.854 | 34.694 | 32.252 |
| Switch-A (Cache$_{c=2}$) | 1.241 | 1.098 | 0.885 | 0.628 | 35.138 | 34.691 | 34.141 | 32.232 |
| Switch-A (Cache$_{c=3}$) | 1.222 | 1.129 | 0.899 | 0.544 | 35.363 | 34.733 | 34.141 | 32.622 |
| Switch-A (Cache$_{c=5}$) | 1.261 | 1.141 | 0.911 | **0.697** | 34.947 | 34.902 | 33.962 | 32.468 |
| CMLoRA (Cache$_{c=2}$) | 0.977 | 1.117 | 0.960 | 0.314 | 35.259 | 34.357 | 34.388 | 33.219 |
| CMLoRA (Cache$_{c=3}$) | 1.090 | 0.986 | 0.829 | 0.035 | 34.773 | 34.180 | 34.081 | 32.948 |
| CMLoRA (Cache$_{c=5}$) | 0.951 | 1.031 | 0.711 | -0.204 | 34.559 | 34.302 | 34.291 | 32.125 |

Table 4: Element-wise Average of ImageReward and ClipScore with the selected LoRA integration methods under different numbers of Anime and Reality LoRAs in the *ComposLoRA* testbed.

| Model | ImageReward | | | | ClipScore | | | |
|---|---|---|---|---|---|---|---|---|
| | N=2 | N=3 | N=4 | N=5 | N=2 | N=3 | N=4 | N=5 |
| Merge (Cache$_D$) | 0.604 | 0.472 | -0.196 | -0.390 | 33.554 | 33.917 | 33.465 | 32.654 |
| Merge | 0.640 | 0.476 | -0.101 | -0.444 | 33.726 | 34.139 | 33.399 | 32.364 |
| Merge (Cache$_{c=2}$) | 0.579 | 0.457 | -0.206 | -0.519 | 32.928 | 33.999 | 32.996 | 31.836 |
| Merge (Cache$_{c=3}$) | 0.558 | 0.414 | -0.228 | -0.591 | 32.887 | 33.596 | 33.326 | 32.004 |
| Merge (Cache$_{c=5}$) | 0.542 | 0.289 | -0.239 | -0.515 | 33.360 | 33.391 | 32.929 | 31.971 |
| Switch (Cache$_D$) | 0.821 | 0.667 | 0.388 | 0.160 | 35.528 | 35.112 | 34.845 | 34.056 |
| Switch | 0.821 | 0.674 | 0.340 | 0.124 | 35.394 | 35.107 | 34.478 | 33.475 |
| Switch (Cache$_{c=2}$) | 0.622 | 0.650 | 0.326 | 0.006 | 34.889 | 34.559 | 34.693 | 33.743 |
| Switch (Cache$_{c=3}$) | 0.720 | 0.602 | 0.313 | 0.107 | 35.182 | 34.968 | 34.670 | 33.742 |
| Switch (Cache$_{c=5}$) | 0.708 | 0.571 | 0.270 | 0.053 | 35.213 | 34.752 | 34.264 | 33.501 |
| Composite (Cache$_D$) | 0.795 | 0.529 | 0.221 | 0.105 | 35.295 | 34.984 | 34.981 | 33.097 |
| Composite | 0.686 | 0.540 | 0.220 | -0.104 | 35.073 | 34.082 | 34.802 | 32.582 |
| Composite (Cache$_{c=2}$) | 0.694 | 0.550 | 0.185 | -0.193 | 34.582 | 34.339 | 34.528 | 32.929 |
| Composite (Cache$_{c=3}$) | 0.663 | 0.513 | 0.175 | -0.311 | 34.394 | 33.904 | 34.595 | 32.353 |
| Composite (Cache$_{c=5}$) | 0.605 | 0.477 | 0.160 | -0.317 | 34.262 | 33.677 | 33.974 | 31.885 |
| LoraHub (Cache$_D$) | 0.740 | 0.617 | 0.256 | 0.053 | 35.609 | 34.919 | 35.135 | 33.659 |
| LoraHub | 0.748 | 0.640 | 0.284 | 0.081 | **35.681** | 35.127 | 34.970 | 33.485 |
| LoraHub (Cache$_{c=2}$) | 0.699 | 0.533 | 0.248 | 0.004 | 35.290 | 34.817 | 34.489 | 32.401 |
| LoraHub (Cache$_{c=3}$) | 0.713 | 0.512 | 0.198 | -0.005 | 35.225 | 34.728 | 34.050 | 32.245 |
| LoraHub (Cache$_{c=5}$) | 0.619 | 0.509 | 0.195 | -0.008 | 35.053 | 34.591 | 34.086 | 31.642 |
| CMLoRA (Cache$_D$) | 0.819 | 0.700 | **0.446** | **0.186** | 35.422 | 35.215 | **35.208** | 34.341 |
| CMLoRA | 0.810 | **0.715** | 0.415 | 0.151 | 35.558 | **35.699** | 35.146 | **34.640** |
| Switch-A (Cache$_D$) | 0.804 | 0.632 | 0.250 | 0.103 | 35.139 | 35.383 | 34.930 | 34.250 |
| Switch-A | 0.671 | 0.662 | 0.303 | 0.052 | 35.451 | 35.383 | 35.177 | 33.366 |
| Switch-A (Cache$_{c=2}$) | 0.760 | 0.662 | 0.322 | 0.105 | 34.802 | 35.140 | 34.718 | 33.818 |
| Switch-A (Cache$_{c=3}$) | 0.804 | -0.236 | 0.316 | 0.055 | 35.332 | 34.911 | 34.796 | 34.026 |
| Switch-A (Cache$_{c=5}$) | **0.872** | 0.125 | 0.342 | 0.102 | 34.992 | 35.137 | 34.736 | 33.766 |
| CMLoRA (Cache$_{c=2}$) | 0.697 | 0.666 | 0.393 | -0.019 | 35.241 | 34.953 | 34.910 | 34.141 |
| CMLoRA (Cache$_{c=3}$) | 0.736 | 0.583 | 0.320 | -0.186 | 34.825 | 34.628 | 34.720 | 33.885 |
| CMLoRA (Cache$_{c=5}$) | 0.670 | 0.614 | 0.225 | -0.303 | 34.499 | 34.864 | 34.516 | 33.174 |

## B.2 COMPUTATIONAL COST ANALYSIS

Across the investigated caching mechanisms, our proposed caching mechanism $Cache_D$ demonstrates the best performance, indicating that multi-LoRA composition methods utilizing $Cache_D$ degrade the semantic accuracy and aesthetic quality of the generated images the least. While the computational cost of the cache mechanism $Cache_D$ lies between that of uniform caching mechanisms $Cache_{c=2}$ and $Cache_{c=3}$, multi-LoRA composition methods with $Cache_D$ outperform those using other uniform caching mechanisms, as shown in Table 4. Notably, $Cache_D$ can achieve, and in some cases surpass, the performance of advanced multi-LoRA composition methods, such as Switch-A and CMLoRA, especially as the number of composed LoRAs $N$ increases. Visual demonstrations of these results are provided in Figures 13 to 16. However, we find that there also exists a trade-off between the performance of multi-LoRA composition methods and their computational cost. Although CMLoRA achieves superior performance compared to Merge and Switch, it comes with higher computational costs. For instance, at $N = 2$, CMLoRA incurs $912.350\,G$ MACs compared to $789.770\,G$ for Merge and $734.053\,G$ for Switch. Similarly, at $N = 5$, CMLoRA reaches $1570.335\,G$ MACs, significantly higher than $946.721\,G$ for Merge and $731.811\,G$ for Switch.

Table 5: Comparison of Multiply-Accumulate Operations (MACs) with the selected LoRA integration methods under different number of LoRAs.

| Model | MACs | | | |
|---|---|---|---|---|
| | N=2 | N=3 | N=4 | N=5 |
| Merge (Cache$_D$) | 481.965 G | 515.452 G | 582.421 G | 599.164 G |
| Merge | 789.770 G | 834.613 G | 924.299 G | 946.721 G |
| Merge (Cache$_{c=2}$) | 525.940 G | 561.047 G | 631.261 G | 648.815 G |
| Merge (Cache$_{c=3}$) | 437.990 G | 469.858 G | 533.582 G | 549.513 G |
| Merge (Cache$_{c=5}$) | 367.642 G | 396.907 G | 455.439 G | 470.071 G |
| Switch (Cache$_D$) | 440.406 G | 438.067 G | 444.335 G | 438.736 G |
| Switch | 734.053 G | 730.914 G | 739.322 G | 731.811 G |
| Switch (Cache$_{c=2}$) | 482.356 G | 479.902 G | 486.476 G | 480.604 G |
| Switch (Cache$_{c=3}$) | 398.457 G | 396.232 G | 402.194 G | 396.868 G |
| Switch (Cache$_{c=5}$) | 331.337 G | 329.295 G | 334.768 G | 329.879 G |
| Composite(Cache$_D$) | 830.898 G | 1341.700 G | 1788.934 G | 2236.167 G |
| Composite | 1401.066 G | 2169.199 G | 2892.266 G | 3615.333 G |
| Composite (Cache$_{c=2}$) | 912.350 G | 1459.914 G | 1946.553 G | 2433.191 G |
| Composite (Cache$_{c=3}$) | 749.445 G | 1223.486 G | 1631.315 G | 2039.143 G |
| Composite (Cache$_{c=5}$) | 619.121 G | 1034.343 G | 1379.124 G | 1723.905 G |
| LoraHub(Cache$_D$) | 481.965 G | 515.452 G | 582.421 G | 599.164 G |
| LoraHub | 789.770 G | 834.613 G | 924.299 G | 946.721 G |
| LoraHub (Cache$_{c=2}$) | 525.940 G | 561.047 G | 631.261 G | 648.815 G |
| LoraHub (Cache$_{c=3}$) | 437.990 G | 469.858 G | 533.582 G | 549.513 G |
| LoraHub (Cache$_{c=5}$) | 367.642 G | 396.907 G | 455.439 G | 470.071 G |
| Switch-A (Cache$_D$) | 440.406 G | 438.067 G | 444.335 G | 438.736 G |
| Switch-A | 734.053 G | 730.914 G | 739.322 G | 731.811 G |
| Switch-A (Cache$_{c=2}$) | 482.356 G | 479.902 G | 486.476 G | 480.604 G |
| Switch-A (Cache$_{c=3}$) | 398.457 G | 396.232 G | 402.194 G | 396.868 G |
| Switch-A (Cache$_{c=5}$) | 331.337 G | 329.295 G | 334.768 G | 329.879 G |
| CMLoRA (Cache$_D$) | 627.265 G | 947.652 G | 1060.289 G | 1272.106 G |
| CMLoRA | 912.350 G | 1223.486 G | 1358.518 G | 1570.335 G |
| CMLoRA (Cache$_{c=2}$) | 667.992 G | 987.057 G | 1102.893 G | 1314.711 G |
| CMLoRA (Cache$_{c=3}$) | 586.539 G | 908.248 G | 1017.685 G | 1229.502 G |
| CMLoRA (Cache$_{c=5}$) | 521.377 G | 845.201 G | 949.519 G | 1161.336 G |

## B.3 MLLM Evaluation

Table 6: Average performance metrics for images generated by different base models, evaluated by MiniCPM across four criteria: Element Integration, Spatial Consistency, Semantic Accuracy, Aesthetic Quality, and their average.

| Model | Element Integration | Spatial Consistency | Semantic Accuracy | Aesthetic Quality | Average |
|---|---|---|---|---|---|
| | | | MiniCPM Evaluation | | |
| Naive | 7.041 | 6.815 | 5.127 | _8.129_ | 6.778 |
| CMLoRA | **7.935** | **7.968** | **7.993** | **8.675** | **8.393** |
| Switch-A | 7.162 | _7.020_ | 6.505 | 7.683 | 7.093 |
| Merge | 6.983 | 6.715 | 6.345 | 7.583 | 6.906 |
| Switch | _7.175_ | 6.995 | 6.533 | 7.735 | _7.110_ |
| Composite | 7.088 | 6.983 | 6.505 | 7.685 | 7.065 |
| LoraHub | 7.135 | 7.007 | _6.543_ | 7.740 | 7.106 |

Table 7: Average performance metrics for different models with our cache mechanism, evaluated by MiniCPM across four criteria: Element Integration, Spatial Consistency, Semantic Accuracy, and Aesthetic Quality, along with their average.

| Model | Element Integration | Spatial Consistency | Semantic Accuracy | Aesthetic Quality | Average |
|---|---|---|---|---|---|
| **MiniCPM Evaluation** | | | | | |
| CMLoRA (Cache$_D$) | **7.955** | **7.973** | **7.978** | **8.665** | **8.143** |
| Switch-A (Cache$_D$) | 7.283 | 7.132 | 6.643 | 7.778 | 7.209 |
| Merge (Cache$_D$) | 6.933 | 6.735 | 6.343 | 7.550 | 6.765 |
| Switch (Cache$_D$) | 7.118 | 6.993 | 6.258 | 7.725 | 6.773 |
| Composite (Cache$_D$) | 7.378 | 7.170 | 6.720 | 7.900 | 7.292 |
| LoraHub (Cache$_D$) | 7.160 | 6.993 | 6.553 | 7.738 | 7.111 |

# C PRESENTATION OF GENERATED IMAGE ACROSS EVALUATED MULTI-LoRA METHODS

## C.1 DEMONSTRATION OF ANIME MULTI-LoRA COMPOSITION

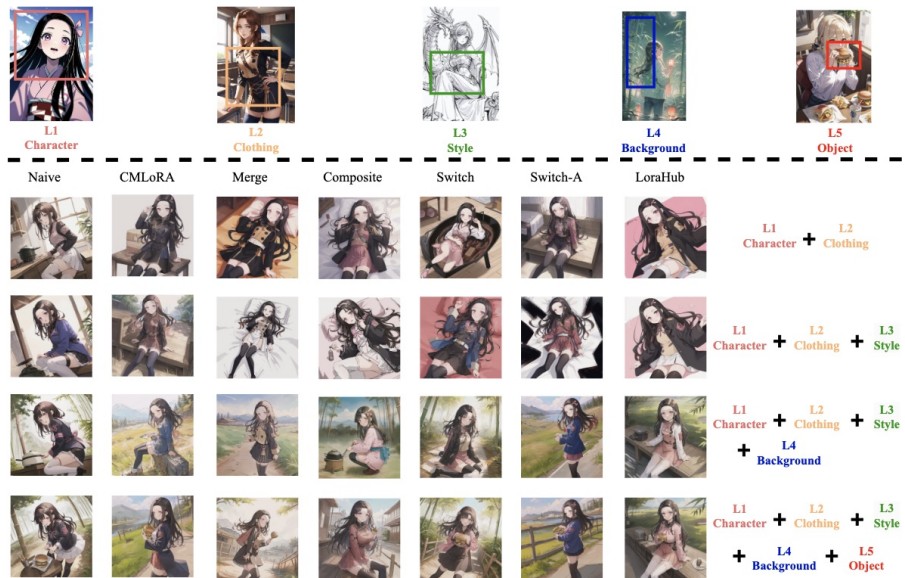

Figure 11: Generated images with different $N$ LoRA candidates ($L1$ Character, $L2$ Clothing, $L3$ Style, $L4$ Background and $L5$ Object) across our proposed framework and baseline methods.

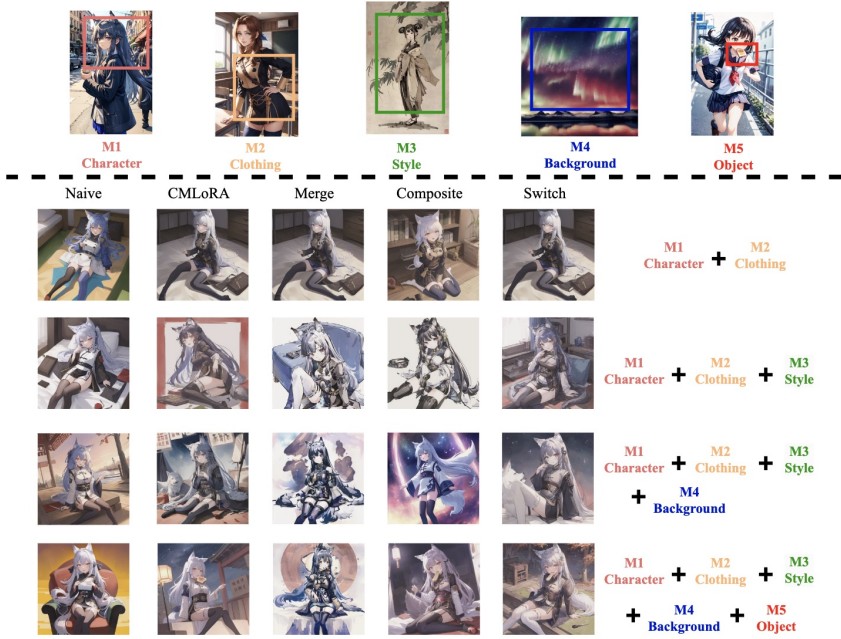

Figure 12: Generated images with different $N$ LoRA candidates ($M1$ Character, $M2$ Clothing, $M3$ Style, $M4$ Background and $M5$ Object) across our proposed framework and baseline methods.

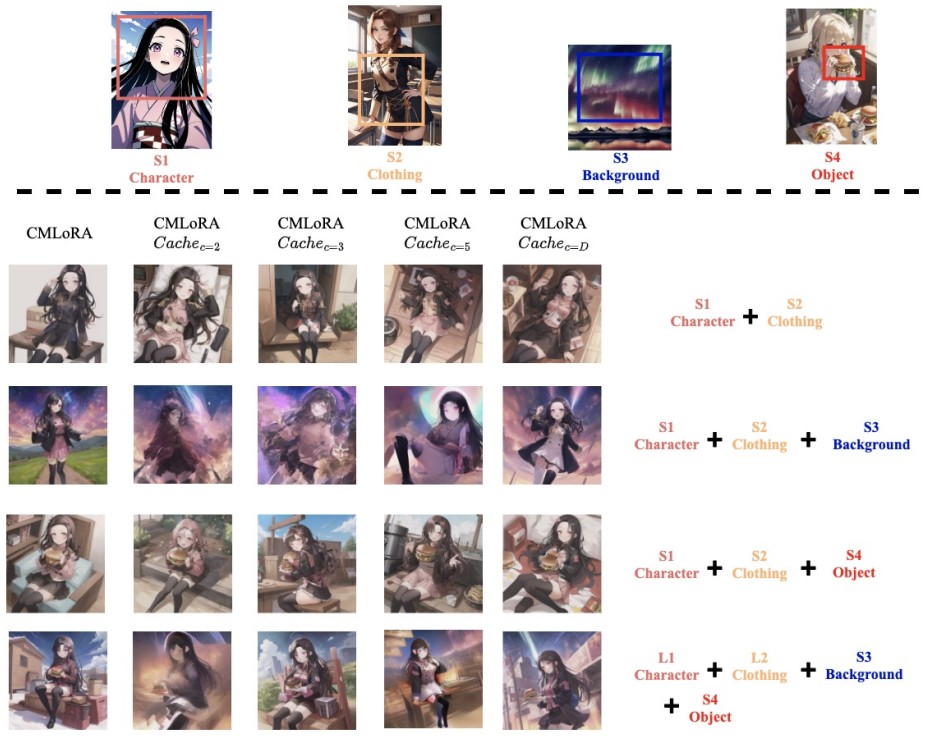

Figure 13: Generated images with different $N$ LoRA candidates ($S1$ Character, $S2$ Clothing, $S3$ Background and $S4$ Object) across CMLoRA with different caching mechanisms.

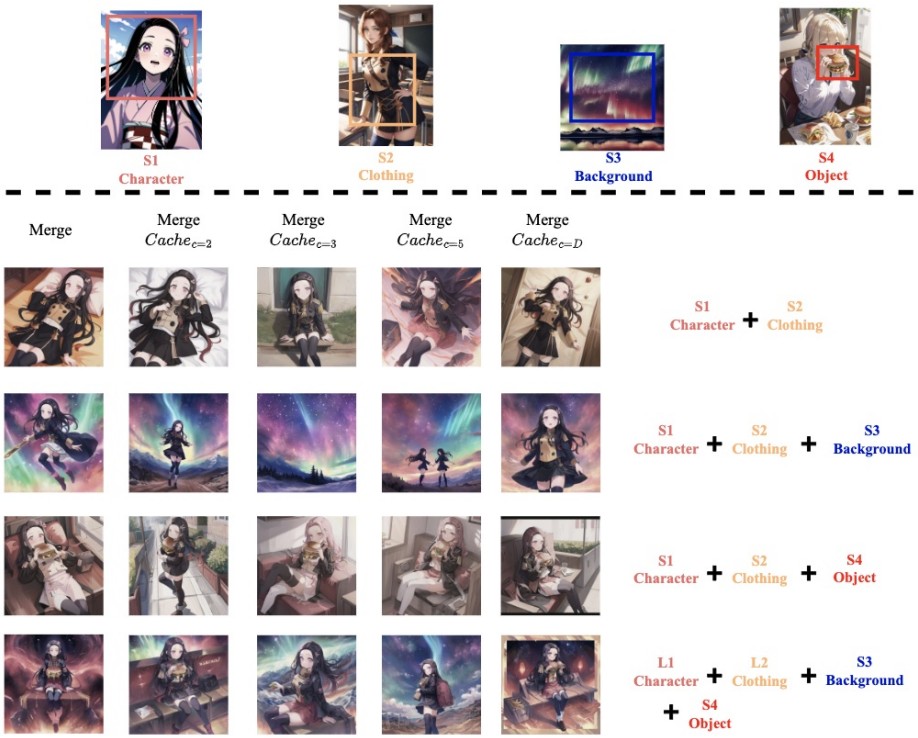

Figure 14: Generated images with different $N$ LoRA candidates ($S1$ Character, $S2$ Clothing, $S3$ Background and $S4$ Object) across Merge with different caching mechanisms.

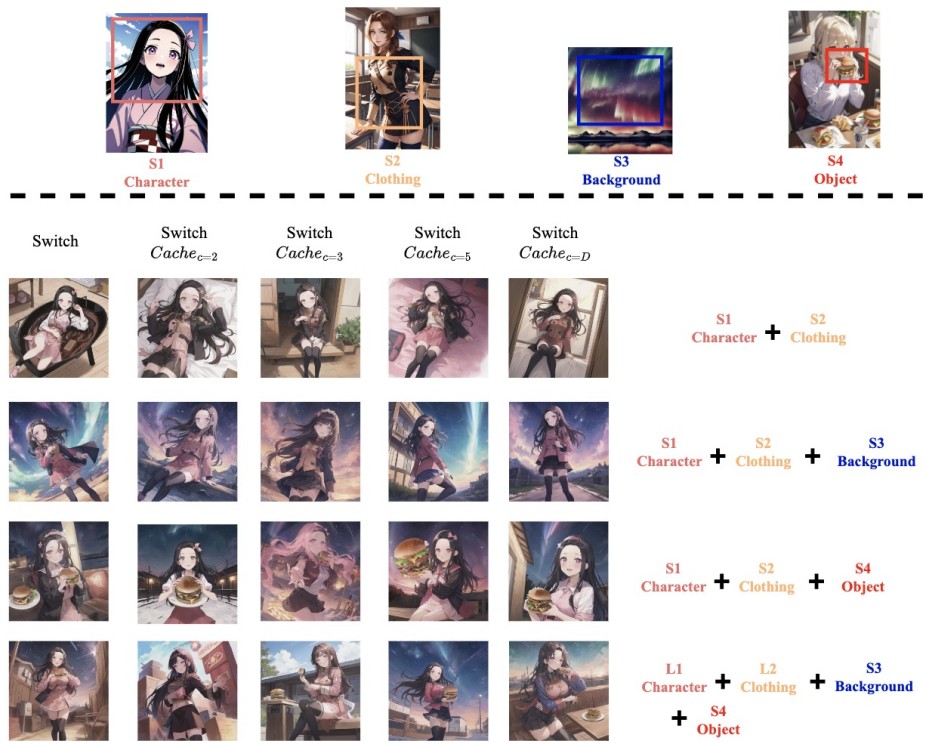

Figure 15: Generated images with different $N$ LoRA candidates ($S1$ Character, $S2$ Clothing, $S3$ Background and $S4$ Object) across Switch with different caching mechanisms.

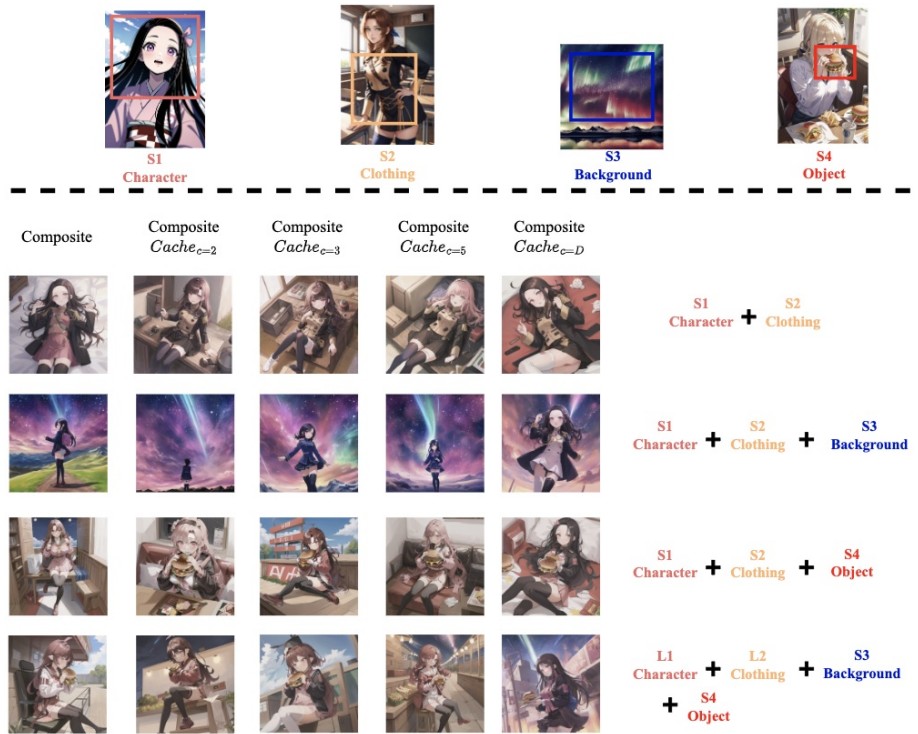

Figure 16: Generated images with different $N$ LoRA candidates ($S1$ Character, $S2$ Clothing, $S3$ Background and $S4$ Object) across Composite with different caching mechanisms.

## C.2 Demonstration of Reality Multi-LoRA Composition

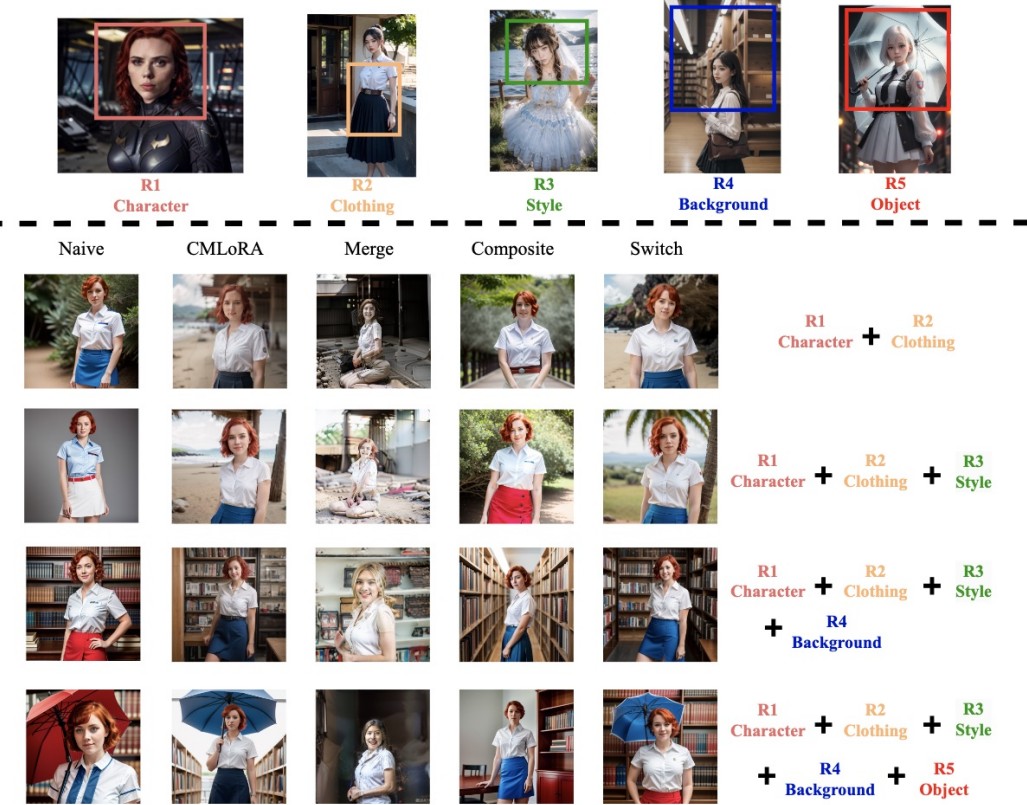

Figure 17: Generated images with different $N$ LoRA candidates ($R1$ Character, $R2$ Clothing, $R3$ Style, $R4$ Background and $R5$ Object) across our proposed framework and baseline methods.

## D LIMITATIONS AND FAILURE CASES

There are some limitations of our work. First, it is important to note that there is a lack of a detailed image generation class taxonomy. This gap poses challenges in systematically classifying well-defined conceptual groups, particularly due to the semantic overlaps that inherently exist among some conceptual categories. These overlaps blur the boundaries between different conceptual categories, making it difficult to establish a robust and well-defined multi-LoRA composition testbed. However, if different LoRA categories possess distinct frequency characteristics, our proposed CM-LoRA approach can still perform effectively. Secondly, as a training-free method, CMLoRA operates independently of additional prior knowledge related to region or layout features, such as bounding box constraints or masked attention maps. This characteristic simplifies its deployment but also introduces certain limitations in handling spatial relationships effectively. As a result, CMLoRA struggles to effectively combine multiple LoRAs within similar semantic categories. This limitation is particularly problematic when multiple concepts within the same conceptual category need to be localized independently. The absence of explicit mechanisms for managing these localizations can lead to potential semantic conflicts, such as concept vanishing or distortion. These issues become especially pronounced when the frequency spectra of overlapping concepts interfere excessively. Finally, we initially employ the traditional image metric, CLIPScore, to evaluate the comprehensive image generation capabilities of all multi-LoRA composition methods. While CLIPScore performs well to evaluate general image-text alignment within its domains, it encounters limitations when applied to scenarios requiring the assessment of out-of-distribution (OOD) concepts, such as user-specific instances. Its evaluations fall short in capturing specific compositional and quality aspects, as it lacks the capability to discern the nuanced features of individual elements (Zhong et al., 2024). This limitation inherently results in a compressed range of evaluation scores for multi-LoRA composition methods, causing improvements to appear marginal despite significant advancements in comprehensive compositional quality. To address this evaluation gap, we leverage the capabilities of multi-modal large language models (MLLMs) to evaluate composable multi-concept image generation in Section 3.2.2.

### D.1 DEMONSTRATION OF FAILURE CASES IN MULTI-LoRA COMPOSITION

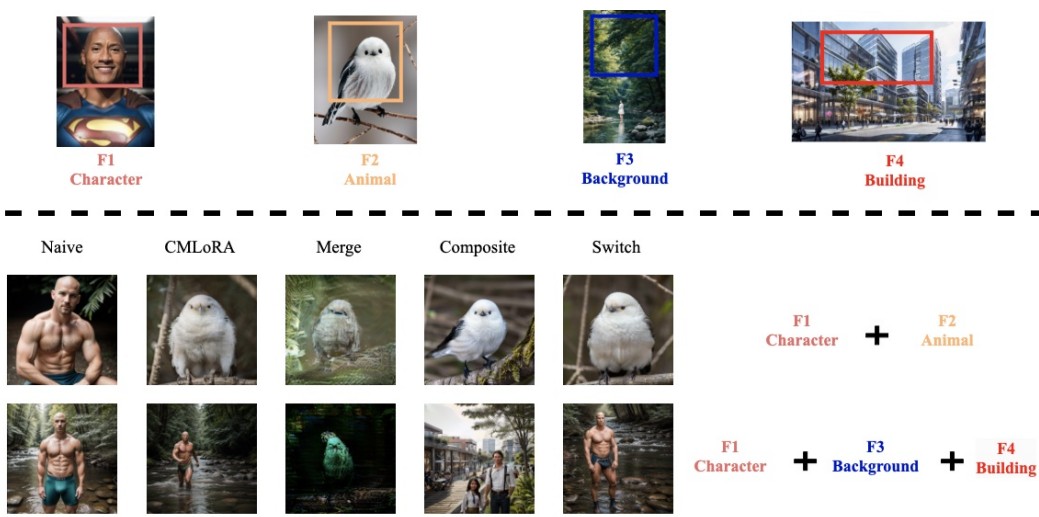

Figure 18: Generated images with different $N$ LoRA candidates ($F1$ Character, $F2$ Animal, $F3$ Background and $F4$ Building) across our proposed framework and baseline methods.

# E    EVALUATION METHODOLOGY

Existing traditional image generation metrics primarily focus on text-image alignment but often overlook the complexity of individual elements within an image and the quality of their composition. Thus, we construct the following evaluation pipeline based on the MLLM.

## E.1    EVALUATION PIPELINE

Image generation: Use both models to generate images based on the same set of prompts (both simple and complex).

In-context few-shot learning: Give a few evaluation examples to the evaluator.

Blind scoring: Let the evaluator rate the images based on the criteria without knowing which model created them.

Score aggregation: Average the scores for each dimension across all prompts to identify overall performance trends.

Comparative analysis: Compare the total and individual dimension scores between models to draw insights on strengths and weaknesses.

## E.2    IMAGE EVALUATION METRICS

### 1) ELEMENT INTEGRATION

Score on a scale of 0 to 10, in 0.5 increments, where 10 is the best and 0 is the worst.

**Description:** How seamlessly different elements are combined within the image.

**Criteria:**

- **Visual Cohesion:** Assess whether elements appear as part of a unified scene rather than disjointed parts.
- **Object Overlap and Interaction:** Check for natural overlaps and interactions between objects, avoiding unnatural placements or intersections.

### 2) SPATIAL CONSISTENCY

Score on a scale of 0 to 10, in 0.5 increments, where 10 is the best and 0 is the worst.

**Description:** Uniformity in style, lighting, and perspective across all elements.

**Criteria:**

- **Stylistic Uniformity:** All elements should share a consistent artistic style (e.g., realism, cartoonish).
- **Lighting and Shadows:** Ensure consistent light sources and shadow directions to maintain realism.
- **Perspective Alignment:** Elements should adhere to a common perspective, avoiding mismatched viewpoints.

### 3) SEMANTIC ACCURACY

Score on a scale of 0 to 10, in 0.5 increments, where 10 is the best and 0 is the worst.

**Description:** Correct interpretation and representation of each element as described in the prompt.

**Criteria:**

- **Object Accuracy:** Objects should match their descriptions in type, attributes, and context.
- **Action and Interaction:** Actions or interactions between objects should be depicted correctly.

4) AESTHETIC QUALITY

Score on a scale of 0 to 10, in 0.5 increments, where 10 is the best and 0 is the worst.

**Description:** Overall visual appeal and artistic quality of the generated image.

**Criteria:**

- **Color Harmony:** Use of color palettes that are visually pleasing and appropriate for the scene.
- **Composition Balance:** Balanced arrangement of elements to create an engaging composition.
- **Clarity and Sharpness:** Images should be clear, with well-defined elements and no unwanted blurriness.

# F   ABLATION ANALYSIS

To enhance our understanding of the proposed methods, we further conduct the following ablation studies in the field of LoRA fusion sequence and caching strategy of non-dominant LoRA based on our proposed CMLoRA.

**Dominant LoRA Order Sequence Determination** Building on our previous discussions, we can identify optimal dominant LoRA candidates during the denoising process, leading us to formulate the following combination optimization problem: *How to derive the denoising range (step-length) of the activated dominant LoRA?*

Utilizing the structure of our Cached Multi-LoRA (CMLoRA) framework, feature maps generated by different LoRAs can be dynamically fused at each inference step. We define the total denoising range (step-length) for a dominant LoRA $i$ as $D_i$. In our configuration, we assume that each LoRA contributes equally, leading us to allocate the dominant range $D_i$ uniformly across all active LoRAs throughout the inference process.

Consider a scenario with a total of $T$ denoising steps and $N$ LoRAs. We set $D_i = \lfloor \frac{T-1}{N} \rfloor, \forall i$. We have high-frequency LoRA set $H$ and low-frequency LoRA set $L$. Inspired by the LoRA switch mechanism, we implement a cyclic pattern of dominant LoRAs among the candidates in set $H$ at the beginning of the denoising process, switching the dominant LoRA every step. To ensure convergence during the denoising process, we designate to use the low-frequency LoRA from the low-frequency LoRA set $L$ at the end of $D_i$ steps. By implementing this approach, we effectively harness the pronounced dynamics of high-frequency components while simultaneously benefiting from the stabilizing attributes of low-frequency elements, ultimately leading to visual consistency of multi-LoRA composition.

**Dominant LoRA Scale** $w_{dom}$    For the dominant LoRA, we assign a weight denoted as $w_{\mathrm{dom}}$. For the non-dominant LoRAs, we set their weights as $w_{\mathrm{non}} = \frac{N}{w_{\mathrm{dom}}+N-1}$, where $N$ is the total number of composed LoRAs. To regulate $w_{\mathrm{dom}}$ during the diffusion process, we employ a decaying method. This decaying strategy not only stabilizes the denoising process but also plays a critical role in reducing semantic conflicts between different LoRAs. By gradually attenuating the influence of the dominant LoRAs as the denoising process progresses, it prevents abrupt changes in texture and edge features that could disrupt with global structures. This ensures smoother transitions between the contributions of various LoRAs, leading to a more harmonious integration of both high- and low-frequency components. As a result, the overall semantic coherence is preserved, minimizing the risk of feature misalignment.

We initially set the dominant weight scale $w_{\mathrm{dom}}$ to $N - \alpha$, where $N$ is the total number of activated LoRAs and $\alpha \in \mathbb{R}^+$. This choice allows us to balance the contribution of the dominant LoRA against the collective contribution of the non-dominant LoRAs. To optimize this balance, we conduct a grid search over $\alpha$ in set $\{0.1, 0.2, \cdots, 0.8, 0.9\}$ By adjusting $\alpha$, we can finetune the influence of the dominant LoRA, ensuring it does not overpower the others. Then we choose the optimal $\alpha = 0.5$.

Based on the principle that *high-frequency components display more pronounced dynamics during the early stage of the denoising process* (Si et al., 2024), this weight scale is adjusted using

a decaying method. For the $i$-th turn of switching the dominant LoRA, the weight is define as:
$w_{\text{dom}}^i = w_{\text{dom}}^{i-1} - 0.5^i$.

**Caching Interval and Modulation Hyper-parameters** $c_1, c_2$   To capture the similarity trend of feature maps fused by LoRAs, we propose a non-uniform caching interval strategy with two specialized hyper-parameters: $c_1, c_2 \in \mathbb{Z}$. These hyper-parameters control the strength of the caching behavior during inference. Specifically, for a denoising process with $T$ timesteps, the sequence of timesteps that performs full inference is:

$$
\begin{aligned}
\mathcal{I} &= \mathcal{I}_1 \cup \mathcal{I}_2 \cup \mathcal{I}_3 \\
\mathcal{I}_1 &= \{c_1 \cdot t \mid 0 \leq c_1 \cdot t < \lfloor 0.4 \cdot T \rfloor, \text{ where } t \in \mathbb{Z}\} \\
\mathcal{I}_2 &= \{\lfloor 0.4 \cdot T \rfloor + c_2 \cdot t \mid \lfloor 0.4 \cdot T \rfloor \leq c_2 \cdot t < \lfloor 0.9 \cdot T \rfloor, \text{ where } t \in \mathbb{Z}\} \\
\mathcal{I}_3 &= \{\lfloor 0.9 \cdot T \rfloor + c_1 \cdot t \mid \lfloor 0.9 \cdot T \rfloor \leq c_1 \cdot t < T, \text{ where } t \in \mathbb{Z}\}.
\end{aligned}
\tag{17}
$$

The discrete interval $\mathcal{I}_4 = [\lfloor 0.4 \cdot T \rfloor, \lfloor 0.9 \cdot T \rfloor] = \{k \in \mathbb{Z} \mid k = 5n, n \in \mathbb{Z}, \lfloor 0.4 \cdot T \rfloor \leq k \leq \lfloor 0.9 \cdot T \rfloor\}$ are established based on the condition that the average similarity $s_t$ of the cached features at timestep $t$ in a interval exceeds $20\%$ within a $90\%$ confidence interval. Since we have only finite discrete samples, we conduct our calculation based on the Monte Carlo method. Formally, $\forall t \in \mathcal{I}_4$, $\overline{\mathcal{P}}(0.2 \in [s_t - \text{std}(s_t), s_t + \text{std}(s_t)]) = 0.9$, where $\overline{\mathcal{P}}$ is the probability averaged on the similarity $s_t$ of the cached features for all discrete timesteps $t \in \mathcal{I}_4$ and std is the standard deviation of $s_t$ at timestep $t$.

Given that cached LoRA features exhibit greater similarity in $\mathcal{I}_4$ compare to $\mathcal{I} \setminus \mathcal{I}_4$, we intuitively select $c_1 < c_2$. This selection is informed by a grid search over the pairs $(c_1, c_2)$ in the Cartesian product of two discrete sets $[1,5] \times [1,5]$. When $c_1 < c_2 < 4$, we find that there is minimal variation in the content of the image, accompanied by only slight fluctuations in the CLIP Score. We observe a performance deterioration if we choose $5 < c_1 < c_2$. Finally, we obtain the optimal caching modulation hyperparameters: $(2, 3)$.

## F.1   Order of LoRA Activation

Zhong et al. (2024) propose that *The initial choice of LoRA in the activation sequence clearly influences overall performance, while alterations in the subsequent order have minimal impact*, so we conduct the following ablation study to demonstrate the effectiveness of our LoRA fusion order based on frequency partition.

Table 8: Comparison of ClipScore with the selected LoRA integration methods under different number of LoRAs.

| Model | Average ClipScore | |
| --- | --- | --- |
| | N=4 | N=5 |
| CMLoRA (Character) | 34.258 | 33.901 |
| CMLoRA (Clothing) | 34.188 | 33.870 |
| CMLoRA (Style) | 34.256 | 33.954 |
| CMLoRA (Background) | 34.224 | 33.807 |
| CMLoRA (Random) | 34.166 | 33.745 |
| CMLoRA | **35.528** | **35.355** |

## F.2   Cache Interval

Additionally, we compare our proposed caching strategy with other uniform caching methods, demonstrating that our approach outperforms them.

Table 9: Comparison of ClipScore with the selected LoRA integration methods under different number of LoRAs.

| Model | ClipScore (N=2) | ClipScore (N=3) | ClipScore (N=4) | ClipScore (N=5) |
|---|---|---|---|---|
| CMLoRA (Cache$_D$) | **35.422** | **35.215** | **35.208** | **34.341** |
| CMLoRA (Cache$_{c=2}$) | 35.241 | 34.953 | 34.910 | 34.141 |
| CMLoRA (Cache$_{c=3}$) | 34.825 | 34.628 | 34.720 | 33.885 |
| CMLoRA (Cache$_{c=5}$) | 34.499 | 34.864 | 34.516 | 33.174 |

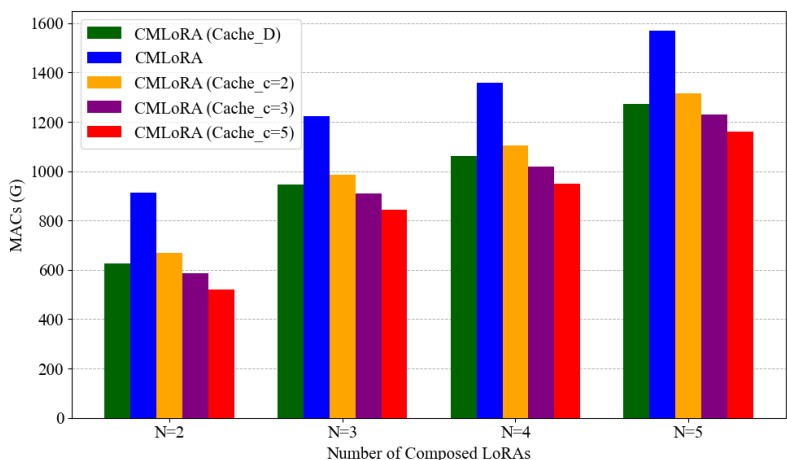

Figure 19: Results of Computational Cost of Different Cache Methods. MACs refer to Multiple-Accumulate Operations.

