# OpenReview forum: "Cached Multi-Lora Composition for Multi-Concept Image Generation"
_ICLR.cc/2025/Conference — ICLR 2025 Poster_

### Official Review · Reviewer_gjCY · 2024-11-04

**Soundness:** 2
**Presentation:** 3
**Contribution:** 2
**Rating:** 5
**Confidence:** 5

**Summary:**

This paper introduces Cached Multi-LoRA (CMLoRA), a framework for training-free, multi-concept image generation that integrates multiple Low-Rank Adaptation (LoRA) modules in text-to-image diffusion models. By analyzing LoRAs in the Fourier domain, CMLoRA partitions LoRAs into high- and low-frequency sets, applying high-frequency LoRAs in early denoising stages and low-frequency ones later to reduce semantic conflicts. A novel caching mechanism selectively activates non-dominant LoRAs, enhancing computational efficiency while maintaining image quality. Evaluated against existing methods, CMLoRA shows superior performance in aesthetic and compositional quality, demonstrating its effectiveness for generating complex, coherent images from multiple concepts.

**Strengths:**

1.  The paper introduces a novel Fourier-based approach to address the challenge of multi-LoRA composition by partitioning LoRA modules into high- and low-frequency categories. This frequency-aware sequencing strategy is innovative, as it moves beyond the typical naive integration of LoRAs by leveraging the frequency domain to systematically order their application during inference. This approach effectively mitigates semantic conflicts and represents a creative combination of LoRA adaptation with Fourier-based analysis, contributing a unique perspective to the field of multi-concept image generation.

2.  The paper’s methodology is sound and well-supported by rigorous experimentation. The introduction of the Cached Multi-LoRA (CMLoRA) framework is methodically detailed, with clear mathematical formulations and a thorough explanation of the caching mechanism. The empirical evaluations are comprehensive, covering a range of established metrics like CLIPScore and MLLM-based benchmarks, which validate the claims across different aspects of multi-concept image synthesis, including element integration, spatial consistency, and aesthetic quality.

3. The proposed CMLoRA framework addresses a significant limitation in current LoRA-based image generation methods by enabling efficient and high-quality integration of multiple LoRA modules. The training-free nature of CMLoRA increases its practical applicability, making it more accessible for scenarios where training resources are limited or infeasible.

**Weaknesses:**

1. What are the failure cases? A couple of visual examples of failed outputs could provide more insights into the limitations of the CMLoRA method.

2. How were the caching hyperparameters $c_1$ and $c_2$ chosen, and how sensitive is the model’s performance to their variations? Furthermore, there is limited discussion of how the caching interval impacts the final performance in terms of both computational efficiency and image quality. Additional experiments that explore the impact of varying these parameters would make the paper’s claims around caching strategy more robust and actionable for readers interested in applying or extending CMLoRA.

3. What is the exact impact of the frequency-based LoRA partitioning, and would alternative sequencing strategies be effective?

4. The paper’s evaluations focus primarily on a limited set of datasets (anime and realistic styles within the ComposLoRA testbed) and may not generalize to broader multi-concept applications. Furthermore, CLIPScore and the other metrics used may not fully capture nuances in compositional fidelity, particularly as the number of LoRAs increases. Expanding the scope of datasets and incorporating additional image quality metrics, such as perceptual quality or domain-specific measures, would strengthen the applicability of CMLoRA across a wider range of practical scenarios.

**Questions:**

1. Could you provide examples or further analysis of cases where CMLoRA might struggle with semantic conflicts? For example, are there certain LoRA combinations or types of images where the method performs suboptimally?

2. Could you clarify how you determined the values for the caching hyperparameters $c_1$ and $c_2$? Did you observe any significant performance variations with different values, and if so, could you provide insights on optimal settings?

3. Have you tested CMLoRA on datasets beyond the anime and realistic styles in the ComposLoRA testbed? If not, could you discuss how the method might adapt to other domains?

---

> ### Author Response · Authors · 2024-11-19
> **Response to Reviewer gjCY (1)**
>
> Thank you for your thoughtful reply, and we are fully aware of your concerns. Please kindly see below for our responses to your comments:
>
> > ### Could you provide examples or further analysis of cases where CMLoRA might struggle with semantic conflicts? For example, are there certain LoRA combinations or types of images where the method performs suboptimally?
>
> Our experiments are conducted under the setting of using a single instance from each category. However, when multiple instances from the same LoRA category (*i.e.*, LoRAs with similar frequency spectra) are introduced during image generation, CMLoRA may encounter challenges with semantic conflicts. For example, as illustrated in Figure 18 of Appendix D.1, combining $F1$ (character) and $F2$ (animal) using CMLoRA may result in a failure of multi-concept composition, leading to a phenomenon known as Concept Vanish.
>
> This limitation reflects a general drawback of existing training-free LoRA composition methods. Without incorporating additional prior knowledge about region or layout features, such as bounding box constraints or masked attention maps, the generative model lacks the capacity to effectively combine multiple LoRAs within similar semantic categories. This limitation is particularly problematic when multiple concepts within the same conceptual category need to be localized independently.
>
> We added a detailed analysis of these limitations, along with visual demonstrations of failure cases, in Appendix D.
>
> > ### Could you clarify how you determined the values for the caching hyperparameters c1 and c2? Did you observe any significant performance variations with different values, and if so, could you provide insights on optimal settings?
>
> We provide details on the selection of hyperparameters in Appendix E: Ablation Analysis. Specifically, we employ a posterior confidence interval check to determine the non-uniform caching interval used during the denoising process. In addition, we select optimal caching modulation hyperparameters $c\_1$ and $c\_2$ based on a grid search method.
>
> Our analysis reveals that when $c\_1<c\_2<4$,  the content of the generated image exhibits minimal variation, with only slight fluctuations observed in the CLIPScore. However, when $5<c\_1<c\_2$, we observe a notable deterioration in performance.

---

> ### Author Response · Authors · 2024-11-19
> **Response to Reviewer gjCY (2)**
>
> > ### Have you tested CMLoRA on datasets beyond the anime and realistic styles in the ComposLoRA testbed? If not, could you discuss how the method might adapt to other domains?
>
> We selected some well-trained LoRAs in Civitai [1], one of the largest available AIGC social platforms, to conduct experiments. We added animals and buildings as new categories to our LoRA test set. Our new testbed includes the following LoRA categories: Character, Cloth, Style, Background, Object, Animal, and Building. However, we find introducing Animal and Building concepts in the multi-LoRA composition will lead to potential semantic conflicts, such as Concept Vanish or Concept Distortion, if we choose multiple concepts in a similar semantic group to compose an image.
>
> We report CLIPScore and average performance metrics for images including animal and building LoRA generated by different multi-LoRA composition methods, evaluated by MiniCPM across four criteria: Element Integration, Spatial Consistency, Semantic Accuracy, Aesthetic Quality, and their average below.
>
> | Model | Average CLIPScore |
> | :-------------------: | :-----------------: |
> | CMLoRA | 33.370 |
> | Switch | 33.076 |
> | Composite | 31.341 |
> | Merge | 28.292 |
>
> | Model | Element Integration | Spatial Consistency | Semantic Accuracy | Aesthetic Quality | Average |
> | :-------------------: | :-----------------: |  :-----------------: | :-----------------: | :-----------------: | :-----------------: |
> | CMLoRA | 7.719 | 7.832 | 5.375 | 8.166 | 7.273 |
> | Switch | 7.063 | 7.042 | 5.235 | 7.182 | 6.631 |
> | Composite | 6.702 | 6.584 | 4.169 | 6.965 | 6.105 |
> | Merge | 4.168 | 5.239 | 3.152 | 3.913 | 4.118 |
>
> Based on our observation, we find that the performance of Semantic Accuracy decreases among all LoRA composition methods, since all methods may omit certain LoRA in the similar semantic group, if we choose multiple concepts in a similar semantic group to compose an image, as shown in Figure 18 in Appendix D.1. LoRA Composite and LoRA Merge deteriorate the most among all LoRA composition methods, since they use the information fused by all LoRAs to compose during the denoising process. We will report all proposed metrics (Clipscore and MiniCPM evaluation score) across investigated multi-LoRA composition methods after running all experiments in our paper.
>
> As highlighted in Appendix D Limitations, a significant issue in the field is the absence of a detailed taxonomy for multi-concept image generation classes. This gap poses challenges in systematically classifying well-defined conceptual groups, particularly due to the semantic overlaps that inherently exist among some conceptual categories. These overlaps blur the boundaries between different conceptual categories, making it difficult to establish a robust and well-defined multi-LoRA composition testbed. However, if different LoRA categories possess distinct frequency spectra characteristics, our proposed CMLoRA approach can still perform effectively. Specifically, we can use the LoRA partition method based on Fourier analysis illustrated in Section 2.2 to profile those LoRA categories and use multiple LoRAs to compose images following the generation pipeline of CMLoRA.
>
> As mentioned in the previous paragraph, we have added our limitation discussion in Appendix D about how the lack of a detailed multi-concept image generation class taxonomy and a well-defined testbed is inherently a limitation.
>
> > ###  What is the exact impact of the frequency-based LoRA partitioning, and would alternative sequencing strategies be effective?
>
> The exact impact of the frequency-based LoRA partitioning is demonstrated through additional visualizations provided in Appendix C. These visualizations compare CMLoRA with multi-LoRA composition methods that do not utilize frequency-based LoRA partitioning. We have also explored alternative sequencing strategies, as discussed in [2], and included the corresponding experimental results in Appendix F.1 Order of LoRA Activation. These results further demonstrate the robustness of CMLoRA.
>
> [1] "The Home of Open-Source Generative AI." Civitai, civitai.com/. Accessed 18 Nov. 2024.
>
> [2] Ming Zhong, Yelong Shen, Shuohang Wang, Yadong Lu, Yizhu Jiao, Siru Ouyang, Donghan Yu, Jiawei Han, and Weizhu Chen. Multi-lora composition for image generation. arXiv preprint arXiv:2402.16843, 2024.

---

> ### Author Response · Authors · 2024-11-19
> **Response to Reviewer gjCY (3)**
>
> >### Furthermore, CLIPScore and the other metrics used may not fully capture nuances in compositional fidelity, particularly as the number of LoRAs increases. Expanding the scope of datasets and incorporating additional image quality metrics, such as perceptual quality or domain-specific measures, would strengthen the applicability of CMLoRA across a wider range of practical scenarios.
>
> Traditional image metrics, while effective for general use cases, have significant shortcomings when applied to scenarios that demand nuanced evaluations of compositional fidelity, especially in out-of-distribution (OOD) contexts. These metrics may compress evaluation ranges and fail to discern the intricate qualities of individual elements in multi-LoRA compositions [2], resulting in marginal performance gains that do not accurately reflect actual advancements.
>
> To address this evaluation gap, we leverage the capabilities of multi-modal large language models (MLLMs) to evaluate composable multi-concept image generation. Using in-context few-shot learning, MLLMs are better equipped to handle challenges posed by OOD samples, offering a more nuanced and context-aware assessment of compositional and quality aspects. This enhanced framework not only addresses the evaluation gap but also ensures a fair and comprehensive validation of the improvements brought by CMLoRA.
>
> We include a detailed explanation in Appendix D Limitations.
>
> We sincerely appreciate it if you could kindly consider improving the scores if we have sufficiently addressed the concerns. We are very happy to answer any further questions you may have.
>
> [2] Ming Zhong, Yelong Shen, Shuohang Wang, Yadong Lu, Yizhu Jiao, Siru Ouyang, Donghan Yu, Jiawei Han, and Weizhu Chen. Multi-lora composition for image generation. arXiv preprint arXiv:2402.16843, 2024.

---

> ### Author Response · Authors · 2024-11-28
>
> Dear Reviewer gjCY,
>
> We appreciate all of the valuable time and effort you have spent reviewing our paper.
>
> As the discussion period concludes in five days, we gently request that you review our reply and consider updating your evaluation accordingly. We believe that we have addressed all questions and concerns raised, but please feel free to ask any clarifying questions you might have before the end of the discussion period.
>
> Authors

---

> ### Author Response · Authors · 2024-12-02
>
> Dear Reviewer gjCY:
>
> We would like to express our sincere gratitude for your valuable time and effort in reviewing our work. We are writing to kindly remind you that the discussion period is drawing to a close.
>
> If you have any remaining questions or concerns about our paper, we would be grateful for the opportunity to address them. We are happy to provide any clarifications you may require.
>
> We fully understand your busy schedule and deeply appreciate your dedication to the review process. Thank you once again for your time.
>
> Authors

---

### Official Review · Reviewer_ok21 · 2024-11-04

**Soundness:** 2
**Presentation:** 3
**Contribution:** 2
**Rating:** 5
**Confidence:** 4

**Summary:**

This paper works on fixing issues of using Lora for multi-concept image generation. Particularly, this paper empirically find that some LoRAs amplify high-frequency features, and others focus on low- frequency elements. Based on this observation, a frequency domain based sequencing strategy is presented to determine the optimal order in which LoRAs should be integrated during inference, and a training-free framework, namely Cached Multi-LoRA (CMLoRA), is designed to integrate multiple LoRAs while maintaining cohesive image generation. Experiments suggest that CMLoRA outperforms SOTA training-free LoRA fusion methods for multi-concept image generation.

**Strengths:**

1.Frequence domain analysis for multi-component generation is indeed an interesting idea.

2.The proposed solution is easy and clear (although high-level insight is not very obvious.)

3.The experiments are good in explain the effectiveness of the solution.

**Weaknesses:**

1.It’s not clear why frequency domain is needed to solve the multi-component generation task. A clear investigation and analysis on how they come up with this solution can further strengthen the contribution of the work. Particularly, more analysis is needed to explain why shift attention from spatial domain to frequency domain.

2.The observation that some LoRAs amplify high-frequency features, and others focus on low- frequency elements is based on a naïve experiment. More analysis or theoretical analysis is needed to better appreciate the proposed idea.

3.The experimental results is good but not convincing to explain the superiority of the solution.

**Questions:**

1.I’m not sure about Figure 1. Do you assume that meaningful amplitude difference happens only at the same time steps for the two Loras? In another word, do you assume different Lora categories are well-aligned along time step? Furthermore, given that the observation in Figure 1 motivates the proposed method, it’s suggested to provide comprehensive analysis to explain the high/low frequency issues of different Loras.

2.The proposed solution in Section 2.2 is presented without deep analysis. Can you please provide a high-level analysis of your solution to explain your method in a progressive way? e.g. eq (2), eq (3) is introduced directly without explain why.

3.The observation is based on Lora categories from Ref1. How does the method perform with respect to different Lora categories?

4.The collective guidance in eq (5) seems related to classifier free guidance, can you please provide further analysis?

5.Benchmark comparison in Table 1 seems marginal performance gain. Please explain further.

6.Please also explain in detail the “semantic conflict” issue as there exists no experiments to verify the existence of this issue (or maybe I failed to find it, please show me where I can find it.)

Ref1, Multi-lora composition for image generation, 2024.

---

> ### Author Response · Authors · 2024-11-19
> **Response to Reviewer ok21 (1)**
>
> Thank you for the thoughtful questions. Please kindly see below for our responses to your comments:
>
> > ### I’m not sure about Figure 1. Do you assume that meaningful amplitude difference happens only at the same time steps for the two Loras? In another word, do you assume different Lora categories are well-aligned along time step? Furthermore, given that the observation in Figure 1 motivates the proposed method, it’s suggested to provide comprehensive analysis to explain the high/low frequency issues of different Loras.
>
> We assume that different LoRA categories exhibit distinct behaviors during the denoising process due to their fusion of varying semantic information into the generated image. Since LoRAs are typically trained independently and fuse features with varying amplitudes across different frequency domains into the generated image, integrating these independent LoRAs with equal contributions to image generation may introduce inherent conflicts.
>
> As discussed in Section 1 Introduction, our motivation stems from an observation in an example: the Character LoRA fuses a higher proportion of high-frequency components, resulting in greater variation in edges and textures compared to the Background LoRA, at the inference stage. This insight forms the basis of our hypothesis that frequency domain scheduling can help multi-concept generation. Specifically, our finding suggests that certain LoRAs introduce more pronounced high-frequency modifications during denoising, whereas others primarily influence low-frequency elements. This can be explained as follows: (1) Some LoRAs enhance high-frequency components, corresponding to rapid changes like edges and textures. (2) Others target low-frequency components, representing broader structures and smooth color transitions.
>
> Furthermore, we observed that high-frequency components of LoRAs are predominantly fused during the early stages of inference, as shown in Figure 3. This observation aligns with prior work showing that high-frequency components vary more significantly than low-frequency ones throughout the denoising process [1]. Consequently, improper integration of various LoRAs may result in visual artifacts or semantic inconsistencies in the generated images.
>
> To make these points clearer, we have edited the text in our introduction and methods sections. Additionally, we have added Figure 2 to the introduction to visually demonstrate how existing methods fail due to these semantic conflicts.
>
> [1] Chenyang Si, Ziqi Huang, Yuming Jiang, and Ziwei Liu. Freeu: Free lunch in diffusion u-net. In Proceedings of the IEEE/CVF Conference on Computer Vision and Pattern Recognition, pp. 4733–4743, 2024.

---

> ### Author Response · Authors · 2024-11-19
> **Response to Reviewer ok21 (2)**
>
> > ### The proposed solution in Section 2.2 is presented without deep analysis. Can you please provide a high-level analysis of your solution to explain your method in a progressive way? e.g. eq (2), eq (3) is introduced directly without explain why.
>
> **Background:** Frequency analysis has proven highly effective in image analysis tasks, such as [1], [2], [3], [4], due to its key advantages:
> 1. **Efficient image feature detection:** Frequency decomposition highlights image features that are often challenging to capture in the spatial domain.
> 2. **Robustness to noise in the spatial domain:** Noise can be effectively isolated and filtered using frequency-based methods.
>
> **Motivation:** Building on these principles, our motivation stems from observations made in Figure 1, where we find that certain LoRAs introduce more pronounced high-frequency modifications during denoising, whereas others primarily influence low-frequency elements. Furthermore, high-frequency components are predominantly fused during the early stages of inference, as confirmed by prior work: high-frequency components vary more significantly than low-frequency ones throughout the denoising process [1].
>
> We assume that different LoRA categories exhibit distinct behaviors during the denoising process, because they fuse features with varying amplitudes across different frequency domains into the generated image. Consequently, improper integration of various LoRAs may result in visual artifacts or semantic inconsistencies in the generated images. Specifically, as shown in Figure 2, we find that directly applying pre-trained LoRA modules to compose the image often leads to semantic conflicts (see LoRA Merge and Switch). This failure primarily arises because independent LoRAs are integrated to contribute equally to image generation during the denoising process.
>
> **Analysis:** Motivated by the phenomenon observed in Figure 1, we propose a Fourier-based method to classify LoRAs with different frequency responses and group them into distinct sets, as shown in Figures 2 and 3. Through our profiling approach, we categorize LoRAs into high-frequency and low-frequency sets. During inference, high-frequency LoRAs are primarily utilized in the early stages of denoising to enhance detail and texture, while low-frequency LoRAs are predominantly applied in the later stages to refine overall structure and coherence.
>
> **Method:** In Equation 2, we first computer the average feature map $\overline{\mathbf{x}}\_{t}$ along the channel dimension $C$ at denoising time $t$. We quantify the amplitude of high-frequency components in the generated image by analyzing its distribution across the frequency spectrum in Equation 3. Then we calculate the change in amplitude of high-frequency components between each time interval during the denoising process in Equation 4.
>
> Based on Equation 4, we can perform the profiling on the LoRA categories in the testbed: 1) establishing a prioritized LoRA order strategy $\mathcal{O}$ using the ranking of variation in the intensity of high-frequency components $\overline{\Delta\mathcal{H}\_{0.2}\left(\overline{\mathbf{x}}\_{t};20\right)}$ across different LoRA categories. 2) Following the strategy $\mathcal{O}$, we can categorize LoRAs into a high-frequency dominant set $H$ and a low-frequency dominant set $L$ for a multi-LoRA composition task. 3) LoRAs from the high-frequency dominant set $H$ are employed predominantly during the initial stages of denoising, where their dynamic features can effectively enhance the image’s detail and texture. In contrast, LoRAs from the low-frequency dominant set $L$ are utilized primarily in the later stages of the denoising process.
>
> To ensure a seamless understanding of our approach, we have expanded the explanations in Section 2, covering the process from categorization and profiling to scheduling.
>
> [1] Chenyang Si, Ziqi Huang, Yuming Jiang, and Ziwei Liu. Freeu: Free lunch in diffusion u-net. In Proceedings of the IEEE/CVF Conference on Computer Vision and Pattern Recognition, pp. 4733–4743, 2024.
>
> [2] Frank, Joel, et al. "Leveraging frequency analysis for deep fake image recognition." International conference on machine learning. PMLR, 2020.
>
> [3] Jiang, Liming, et al. "Focal frequency loss for image reconstruction and synthesis." Proceedings of the IEEE/CVF international conference on computer vision. 2021.
>
> [4] Li, Jia, et al. "Finding the secret of image saliency in the frequency domain." IEEE transactions on pattern analysis and machine intelligence 37.12 (2015): 2428-2440.

---

> ### Author Response · Authors · 2024-11-19
> **Response to Reviewer ok21 (3)**
>
> > ### The observation is based on Lora categories from Ref1. How does the method perform with respect to different Lora categories?
>
> We selected some well-trained LoRAs in Civitai [5], one of the largest available AIGC social platforms, to conduct experiments. We added animals and buildings as new categories to our LoRA test set. Our new testbed includes the following LoRA categories: Character, Cloth, Style, Background, Object, Animal, and Building. However, we find introducing Animal and Building concepts in the multi-LoRA composition will lead to potential semantic conflicts, such as Concept Vanish or Concept Distortion, if we choose multiple concepts in a similar semantic group to compose an image.
>
> We report CLIPScore and average performance metrics for images including animal and building LoRA generated by different multi-LoRA composition methods, evaluated by MiniCPM across four criteria: Element Integration, Spatial Consistency, Semantic Accuracy, Aesthetic Quality, and their average below.
>
> | Model | Average CLIPScore |
> | :-------------------: | :-----------------: |
> | CMLoRA | 33.370 |
> | Switch | 33.076 |
> | Composite | 31.341 |
> | Merge | 28.292 |
>
> | Model | Element Integration | Spatial Consistency | Semantic Accuracy | Aesthetic Quality | Average |
> | :-------------------: | :-----------------: |  :-----------------: | :-----------------: | :-----------------: | :-----------------: |
> | CMLoRA | 7.719 | 7.832 | 5.375 | 8.166 | 7.273 |
> | Switch | 7.063 | 7.042 | 5.235 | 7.182 | 6.631 |
> | Composite | 6.702 | 6.584 | 4.169 | 6.965 | 6.105 |
> | Merge | 4.168 | 5.239 | 3.152 | 3.913 | 4.118 |
>
> Based on our observation, we find that the performance of Semantic Accuracy decreases among all LoRA composition methods, since all methods may omit certain LoRA in the similar semantic group, if we choose multiple concepts in a similar semantic group to compose an image, as shown in Figure 18 in Appendix D.1. LoRA Composite and LoRA Merge deteriorate the most among all LoRA composition methods, since they use the information fused by all LoRAs to compose during the denoising process. We will report all proposed metrics (Clipscore and MiniCPM evaluation score) across investigated multi-LoRA composition methods after running all experiments in our paper.
>
> As highlighted in Appendix D Limitations, a significant issue in the field is the absence of a detailed taxonomy for multi-concept image generation classes. This gap poses challenges in systematically classifying well-defined conceptual groups, particularly due to the semantic overlaps that inherently exist among some conceptual categories. These overlaps blur the boundaries between different conceptual categories, making it difficult to establish a robust and well-defined multi-LoRA composition testbed. However, if different LoRA categories possess distinct frequency spectra characteristics, our proposed CMLoRA approach can still perform effectively. Specifically, we can use the LoRA partition method based on Fourier analysis illustrated in Section 2.2 to profile those LoRA categories and use multiple LoRAs to compose images following the generation pipeline of CMLoRA.
>
> As mentioned in the previous paragraph, we have added our limitation discussion in Appendix D about how the lack of a detailed multi-concept image generation class taxonomy and a well-defined testbed is inherently a limitation.
>
> > ### The collective guidance in eq (5) seems related to classifier free guidance, can you please provide further analysis?
>
> Each element in the collective guidance functions as classifier-free guidance corresponding to the generative model with a single conceptual LoRA. By applying weighted summation to these elements, CMLoRA ensures harmonized guidance throughout the image generation process, enabling the cohesive integration of all elements represented by the different LoRAs.
>
> In response to the reviewer's suggestions, we have added Section A.4.1 in the Appendix to illustrate the relationship between CMLoRA and classifier-free guidance, providing further clarification of their relevance.
>
> [5] "The Home of Open-Source Generative AI." Civitai, civitai.com/. Accessed 18 Nov. 2024.

---

> ### Author Response · Authors · 2024-11-19
> **Response to Reviewer ok21 (4)**
>
> > ### Benchmark comparison in Table 1 seems marginal performance gain. Please explain further.
>
> The seemingly marginal performance gain observed in Table 1 arises primarily from the limitations of the traditional image evaluation metric, CLIPScore, which we initially used to benchmark multi-LoRA composition methods. While CLIPScore is effective in evaluating general image-text alignment within its domains, it has significant shortcomings when applied to scenarios requiring the assessment of out-of-distribution (OOD) concepts, such as user-specific instances. Its evaluations may fall short in capturing specific compositional and quality aspects, as it lacks the capability to discern the nuanced features of individual elements [6]. This limitation inherently results in a compressed range of evaluation scores for multi-LoRA composition methods, causing improvements to appear marginal despite significant advancements in comprehensive compositional quality.
>
> To address this evaluation gap, we leverage the capabilities of multi-modal large language models (MLLMs) to evaluate composable multi-concept image generation. Using in-context few-shot learning, MLLMs are better equipped to handle challenges posed by OOD samples, offering a more nuanced and context-aware assessment of compositional and quality aspects. This enhanced framework not only addresses the evaluation gap but also ensures a fair and comprehensive validation of the improvements brought by CMLoRA.
>
> We include a detailed explanation in Appendix D Limitations.
>
> > ### Please also explain in detail the “semantic conflict” issue as there exists no experiments to verify the existence of this issue (or maybe I failed to find it, please show me where I can find it.)
>
> We added Figure 2 to demonstrate three types of semantic conflicts raised in multi-LoRA composition methods. In the spatial domain, potential semantic conflict of multi-concept composition may be: 1) Concept Misalignment: some concepts are generated with false semantic information 2) Concept Vanish: some concepts may be completely ignored 3) Concept Distortion: some concepts are incorrectly combined.
>
> The first case may arise from insufficient semantic information from a specific LoRA being fused into the generated image. The second scenario may result in a dominance issue, where one LoRA model overshadows the contributions of others, causing the generation process to lean heavily towards its specific attributes or style, thereby failing to generate a balanced representation. In the third case, multiple content-specific LoRAs where features intended to represent different subjects blend indistinctly, leading to a loss of integrity and recognizability for each concept.
>
> We attribute the semantic conflict to the frequency discordance of multi-LoRA composition during the denoising process, since LoRAs are typically trained independently and fuse features across different frequency domains to the generated image.
>
> We also include additional visual examples of semantic conflict in Appendix D.
>
> We hope this time our additional results can sufficiently resolve your concerns. We sincerely appreciate it if you could kindly consider improving the score, and are very happy to answer any further questions you may have.
>
> [6] Ming Zhong, Yelong Shen, Shuohang Wang, Yadong Lu, Yizhu Jiao, Siru Ouyang, Donghan Yu, Jiawei Han, and Weizhu Chen. Multi-lora composition for image generation. arXiv preprint arXiv:2402.16843, 2024.

---

> ### Author Response · Authors · 2024-11-28
>
> Dear Reviewer ok21:
>
> Thank you for your time and effort in reviewing our paper.
>
> We firmly believe that our response and revisions can fully address your concerns. We are open to discussion if you have any additional questions or concerns, and if not, we kindly ask you to reevaluate your score.
>
> Thank you again for your reviews which helped to improve our paper!
>
> Authors

---

> ### Author Response · Authors · 2024-12-02
>
> Dear Reviewer ok21:
>
> We would like to express our sincere gratitude for your valuable time and effort in reviewing our work. We are writing to kindly remind you that the discussion period is drawing to a close.
>
> If you have any remaining questions or concerns about our paper, we would be grateful for the opportunity to address them. We are happy to provide any clarifications you may require.
>
> We fully understand your busy schedule and deeply appreciate your dedication to the review process. Thank you once again for your time.
>
> Authors

---

### Official Review · Reviewer_pnTL · 2024-11-09

**Soundness:** 2
**Presentation:** 3
**Contribution:** 2
**Rating:** 6
**Confidence:** 5

**Summary:**

In this paper, the authors propose an analysis of typical LoRA algorithms when subjected to a caching mechanism. The study is further extended with the proposal of a framework integrating multiple LoRA mechanisms, aiming at reducing concept-related uncertainty, which is expected to show reduced semantic misconceptions. The proposed method is extensively evaluated in terms of CLIPScore and  MiniCPM-V  testing.

**Strengths:**

The paper is generally well written and, (at least for the class of similar papers) rather easy to follow.
The claims of the authors, on which the paper writing discourse is based on, are verified through evaluations which can become clear, if correctly exemplified.

**Weaknesses:**

Even if the writing is good, the quality of the visuals (e.g Fig 4, 6) can be improved.
A lack of visual comparisons is not expected, given the fact that the most of the evaluations showing a certain advantage of the proposed method are either purely subjective or extremely difficult to quantify.
At least in terms of quantitative evaluations (in terms of CLIPScore), the introduction of the cache mechanism does not show consistent results, but rather mixed. A systemic improvement/degradation of the performance its difficult to identify or explain, at least for the cache mechanism analysis.
A total lack of evaluations in terms of computational effort/efficiency.

**Questions:**

Does a multiple LoRA mechanisms ensemble improve the behavior of the generative model in terms of concepts that are under-represented at the data level?
Can you provide some examples in which the method shows improved semantic consistency?
Why are the claims at the end of page 9 and the beginning of page 10 not proven through a visual comparison?
What (or how can be quantified) is the computational effort of the compared methods?

**Details Of Ethics Concerns:**

It's unlikely that this work has more potential to generate harmful images than the previous published work.

---

> ### Author Response · Authors · 2024-11-19
> **Response to Reviewer pnTL (1)**
>
> Thank you for your detailed review and insightful comments. Please kindly see below for our responses to your comments:
>
> > ### Does a multiple LoRA mechanisms ensemble improve the behavior of the generative model in terms of concepts that are under-represented at the data level?
>
> A multiple LoRA mechanism can generally enhance the generative model's behavior, especially for concepts that are under-represented in the training data. This is because LoRA can inject prior knowledge about specific instances into the model, improving its ability to generate relevant outputs for those concepts.
>
> To validate this claim, we conduct empirical experiments. However, since we do not have access to the training data of our backbone generative model, Stable Diffusion v1.5, we employ a posterior method to identify combinations of multiple concepts that may be under-represented at the data level.
>
> We first define the dataset threshold being the average MiniCPM score when running the generation without LoRAs (with only text prompts, we call this Naive model) over the whole test dataset. We then define under-represented concepts as combinations of different LoRA categories that, when fused into the generative model, result in MiniCPM evaluation scores that fall below the dataset threshold, practically this means when their MiniCPM scores are below $6.778$.
>
> We report CLIPScore and average MLLM evaluation metrics for images that only include these under-represented concepts in the two tables below. We compare images generated by the CMLoRA framework, alongside those produced by the naive backbone model without LoRA fusion (using the same text prompt as a condition). These images are evaluated by MiniCPM across four criteria: Element Integration, Spatial Consistency, Semantic Accuracy, Aesthetic Quality, and their overall average score.
> | Model | Average CLIPScore |
> | :-------------------: | :-----------------: |
> | Naive Model | 33.8494 |
> | CMLoRA | 34.2665 |
>
> | Model | Element Integration | Spatial Consistency | Semantic Accuracy | Aesthetic Quality | Average |
> | :-------------------: | :-----------------: |  :-----------------: | :-----------------: | :-----------------: | :-----------------: |
> | Naive Model | 6.721 | 6.582 | 4.150 | 7.887 | 6.335 |
> | CMLoRA | 7.826 | 7.715 | 7.742 | 8.516 | 7.950 |
>
> The quantitative results imply that CMLoRA outperforms the Naive Model across all metrics. In terms of CLIPScore, which reflects concept alignment between generated images and textual prompts, CMLoRA scores higher (34.2665) than the Naive Model (33.8494). When evaluated on MLLM dimensions, CMLoRA demonstrates substantial improvements: it achieves higher ratings in Element Integration (7.826 vs. 6.721), Spatial Consistency (7.715 vs. 6.582), Semantic Accuracy (7.742 vs. 4.150), and Aesthetic Quality (8.516 vs. 7.887). These results, particularly the notable increase in the Semantic Accuracy score, suggest that CMLoRA provides better semantic coherence and visual quality for multi-concept image generation tasks, including concepts that are under-represented at the data level, when compared to the Naive Model.
>
> In addition, we have added the proposed metrics (CLIPScore and MiniCPM evaluation scores) of the Naive Model for multi-concept generation within our testbed in Table 1 and Table 6. We also include visualizations in Figures 11–12 and Figure 17. These findings demonstrate that utilizing a multiple LoRA mechanism ensemble can enhance the performance of the generative model, particularly in improving Semantic Accuracy.
>
> > ### Can you provide some examples in which the method shows improved semantic consistency?
>
> We presented our quantitative results in Table 1 and Figures 7–8. We also added some qualitative results, including real-generation examples of anime and reality multi-LoRA compositions for multi-concept image generation, in Figures 11–12 and Figure 17 in Appendix C. These results demonstrate that our proposed CMLoRA effectively mitigates semantic conflicts, such as concept misalignment and concept distortion, in multi-LoRA compositions. This improvement is achieved through its frequency-domain-based LoRA scheduling mechanism, which ensures more coherent and aligned concept integration.

---

> ### Author Response · Authors · 2024-11-19
> **Response to Reviewer pnTL (2)**
>
> > ### Why are the claims at the end of page 9 and the beginning of page 10 not proven through a visual comparison?
>
> Due to space constraints, we focused on presenting key results in the main text through a radar map (Figure 7) and win rate figures (Figure 8) to support our first claim, and Figures 9–10 to substantiate our second claim.
>
> In response to the reviewer’s suggestion, we have expanded our visual analysis in Appendix C to strengthen our claims further.
>
> **For the first claim**, we added Figures 11–12 and Figure 17 in Appendix C. These figures validate that CMLoRA demonstrates superior performance compared to other multi-LoRA fusion methods in resolving semantic conflicts during multi-LoRA composition. They highlight CMLoRA’s effectiveness in addressing challenges such as misalignment and distortion, ensuring cohesive multi-concept integration.
>
> **For the second claim**, we introduced Figures 13–16 in Appendix C. These figures provide visual comparisons of generated images across varying numbers of LoRA candidates ($S1$: Character, $S2$: Clothing, $S3$: Background, and $S4$: Object) and demonstrate how our proposed caching mechanism ($Cache_{D}$) enhances conceptual LoRA performance. The figures illustrate that as the number of LoRAs $N$ increases, $Cache_{D}$ becomes increasingly critical in optimizing composition outcomes and improving overall performance.
>
> These additional figures provide comprehensive visual evidence that addresses the reviewer’s concern and further supports the claims made in the paper.
>
> > ### What (or how can be quantified) is the computational effort of the compared methods?
>
> We provide a detailed analysis of the computational cost associated with the investigated multi-LoRA composition methods in Appendix B.2. The results, summarized in Table 5, present the computational cost for all methods, measured in terms of Multiply-Accumulate Operations (MACs). Additionally, we specifically visualize the computational cost of CMLoRA in Figure 11 in Appendix E.2, offering further insights into its efficiency.
>
> > ### At least in terms of quantitative evaluations (in terms of CLIPScore), the introduction of the cache mechanism does not show consistent results, but rather mixed. A systemic improvement/degradation of the performance its difficult to identify or explain, at least for the cache mechanism analysis.
>
> While traditional metrics like CLIPScore are widely used for general evaluations, they exhibit notable limitations in scenarios requiring nuanced assessments, such as compositional fidelity in out-of-distribution (OOD) contexts. These metrics may compress evaluation ranges and fail to discern the intricate qualities of individual elements in multi-LoRA compositions, resulting in marginal performance gains that do not accurately reflect actual advancements.
>
> To provide further clarity regarding the cache mechanism analysis, we have expanded our discussion in Appendix B.2 Computational Cost Analysis. The key findings indicate that while the computational cost of the cache mechanism ($Cache_{D}$) is positioned between that of uniform caching mechanisms $Cache_{c=2}$ and $Cache_{c=3}$, multi-LoRA composition methods utilizing $Cache_{D}$ consistently outperform those employing other uniform caching strategies. This highlights the effectiveness of $Cache_{D}$ in balancing computational efficiency with improved compositional fidelity.
>
> We sincerely appreciate it if you could kindly consider improving the scores if the above response can sufficiently address your concerns. We are very happy to answer any further questions you may have.

---

> > ### Comment · Reviewer_pnTL · 2024-11-19
> > **Computational cost overview**
> >
> > So, just to summarize, when comparing Tab. 1 with Tab.5, for N=5 the method achieves an advantage (in terms of CLIP Score), over SwitchA of 0.091 with 839 GMACs more in computation. Similarly, for N=4, the difference between CMLoRA and LoRA Hub (see the typo in the main paper!!) is quantified as an advantage of 0.073 in CLIPScore at a disadvantage of 434 more GMACs, or for N=3, CMLoRA exhibits a disadvantage with a deficit of 0.168 CLIPScore with also 493 more GMACS. The question is how one can prove that the additional computational cost is needed, or what represents a marginal improvement over the current SOTA, under which such an added computational cost could not bejustified?  (with the obvious question of how much one can trust the CLIPScore in this comparison!)

---

> > > ### Author Response · Authors · 2024-11-19
> > > **Response to Reviewer pnTL**
> > >
> > > The primary contribution of our proposed CMLoRA lies in multi-concept image generation, particularly for scenarios with $N>2$ concepts. While CLIPScore is widely reported as a traditional metric in image generation literature, it is not well-suited for evaluating images with multiple user-specific concepts. This limitation is why we emphasize Figure 8, which offers a more robust and fair comparison of CMLoRA ($\text{Cache}\_{D}$) against other multi-LoRA composition methods based on MLLM evaluation win rates. With the proposed caching mechanism, CMLoRA achieves average win rates of $57.7$% against LoRA Hub and $60.4$% against Switch-A across $N=2,3,4,5$, clearly demonstrating its effectiveness despite the higher computational cost. Additionally, Tables 6 and 7 provide detailed MLLM evaluation scores for all investigated Multi-LoRA composition methods, offering comprehensive support for this analysis.
> > >
> > > We corrected the minor typo error in our main paper.

---

### Author Response · Authors · 2024-11-19
**Summary of Revisions**

We thank all the reviewers for their valuable feedback and insightful suggestions. Based on the reviews, we have made the following revisions to our paper:

1. We provide additional visual comparisons of generated images across varying numbers of $N$ LoRA candidates. (Appendix C) [Reviewer pnTL, ok21, gjCY]

2. We expand our analysis to elucidate the motivation behind shifting attention from the spatial domain to the frequency domain, with a comprehensive discussion. (Section 1) [Reviewer ok21]

3. We enhance the clarity of our method by presenting detailed explanations in a progressive way. (Section 2) [Reviewer ok21]

4. We extend the scope of our study to consider a wider range of multi-concept applications, incorporating additional LoRA categories and expanding the meta LoRA categories within the ComposLoRA testbed. (Appendices C, D) [Reviewer ok21, gjCY]

5. We provide a thorough explanation of the computational costs associated with the investigated multi-LoRA composition methods under investigation. (Appendix B.2) [Reviewer pnTL, ok21, gjCY]

6. We introduce a new section highlighting the limitations of our proposed CMLoRA framework and discussing failure cases in multi-concept generation. (Appendix D) [Reviewer pnTL]

7. We have improved the quality of visual illustrations (e.g., Figures 4 and 6) and addressed minor typo errors throughout our work. [Reviewer pnTL]

We have also addressed each reviewer’s comments with more detailed, in-depth responses. Once again we appreciate all the suggestions made by reviewers to improve our work. It is our pleasure to hear your feedback, and we look forward to answering your follow-up questions.

---

### Meta-Review · Area_Chair_GvWF · 2024-12-16

**Metareview:**

The paper was reviewed by three experts and, initially, they provided unanimous "5: marginally below the acceptance threshold".

The authors provided responses to reviewers' raised concerns and new results.

Only Reviewer pnTL participated in the discussion with the authors and concluded that "The proposed method clearly presents some advantages over the current SOTA in terms of various aspects regarding image generation quality. However, this comes at a price, obvious in terms of computational cost."

Reviewers ok21 and gjCY did not react to the authors' responses, and did not provide final assessments.

The AC carefully checked the paper, the reviews and the responses and agrees with Reviewer pnTL that the proposed method clearly has merits in comparison with the current state-of-the-art but is computationally heavy, and that most of the Reviewers ok21 and gjCY raised concerns were addressed by the authors.
All-in-all, it is an interesting work with good results and of interest for the image generation community.

Therefore the AC recommends acceptance.

**Additional Comments On Reviewer Discussion:**

Only Reviewer pnTL participated in the discussion with the authors and concluded that "The proposed method clearly presents some advantages over the current SOTA in terms of various aspects regarding image generation quality. However, this comes at a price, obvious in terms of computational cost."

The other two reviewers did not provide a final, post-rebuttal, assessment, nor interacted with the authors despite the fact that the authors provided early responses.

---

### Decision · Program_Chairs · 2025-01-22

Accept (Poster)